# Role of neutrophil extracellular traps in radiation resistance of invasive bladder cancer

Surashri Shinde-Jadhav[1,2], Jose Joao Mansure[1], Roni F. Rayes[3,4], Gautier Marcq [1], Mina Ayoub[1,2], Rodrigo Skowronski[1], Ronald Kool [1,2], France Bourdeau[3], Fadi Brimo[5], Jonathan Spicer [2,3,4,6 ✉] & Wassim Kassouf [1,2,6 ✉]

Radiation therapy (RT) is used in the management of several cancers; however, tumor radioresistance remains a challenge. Polymorphonuclear neutrophils (PMNs) are recruited to the tumor immune microenvironment (TIME) post-RT and can facilitate tumor progression by forming neutrophil extracellular traps (NETs). Here, we demonstrate a role for NETs as players in tumor radioresistance. Using a syngeneic bladder cancer model, increased NET deposition is observed in the TIME of mice treated with RT and inhibition of NETs improves overall radiation response. In vitro, the protein HMGB1 promotes NET formation through a TLR4-dependent manner and in vivo, inhibition of both HMGB1 and NETs significantly delays tumor growth. Finally, NETs are observed in bladder tumors of patients who did not respond to RT and had persistent disease post-RT, wherein a high tumoral PMN-to-CD8 ratio is associated with worse overall survival. Together, these findings identify NETs as a potential therapeutic target to increase radiation efficacy.

[1] Urologic Oncology Research Division, McGill University Health Centre, Montreal, Canada. [2] Department of Surgery, Faculty of Medicine, McGill University, Montreal, Canada. [3] Division of Thoracic Surgery, McGill University Health Centre, Montreal, Canada. [4] Goodman Cancer Research Center, McGill University, Montreal, Canada. [5] Department of Pathology, McGill University Health Center, Montreal, Canada. [6] These authors jointly supervised this work: Jonathan Spicer, Wassim Kassouf. ✉email: jonathan.spicer@mcgill.ca; wassim.kassouf@muhc.mcgill.ca

Bladder cancer is the 10th most common cause of cancer related deaths worldwide and 25% of newly diagnosed patients present with muscle-invasive disease[1]. Although radical cystectomy is a standard surgical treatment for the management of muscle-invasive bladder cancer (MIBC), the 5-year survival is only 50%[2,3]. In addition to having a major impact on quality of life, surgery is not feasible in over 40% of patients, and many will succumb to the disease without receiving any definitive therapy[4,5]. Bladder-sparing treatment regimens such as radiation therapy (RT) offer an opportunity to fulfill this unmet need as, it has shown favorable oncological outcomes while maintaining quality of life[6]. Meanwhile, radioresistance of tumor cells remains an impediment for effective treatment as, up to 30% of patients have radioresistant tumors and require salvage cystectomy[7].

Intrinsic mechanisms of tumor radioresistance such as DNA damage repair, autophagy and cell-cycle arrest have been well characterized[8]. Emerging evidence indicates that the tumor immune microenvironment (TIME) plays a pivotal role in therapy resistance[9]. Evasion of immune destruction and inflammation are hallmarks of cancer, and key features of the altered TIME post-RT[10,11]. Decreased infiltration and activation of effector cells, increased infiltration of immunosuppressive cells, and the release of pro-inflammatory cytokines can sustain immunosuppression within the TIME, contributing to tumor radioresistance[12]. Gaining a deeper understanding of the extrinsic influence of the TIME, offers the opportunity to improve patient outcomes by increasing the efficacy of RT, and selecting patients that will benefit from this treatment.

Radiation induced inflammation causes a rapid influx of polymorphonuclear neutrophils (PMNs) to the TIME[13]. PMNs are first responders in the innate immune system and represent the most abundant (50–70%) circulating leukocytes in humans[14]. In the context of cancer, a paradoxical role of PMNs has been reported through tumor-induced granulopoiesis[15], increased angiogenesis[16] and PMN polarization[17]. Clinically, elevated circulating PMNs have been linked to adverse prognosis in patients with various cancer types including bladder cancer[18–20]. This has also been associated with poor local control in patients treated with radiation-based therapy[21]. A high PMN infiltrate is observed in the TIME of bladder cancer among other cancer types, where histological analysis shows an association between high intratumoral PMN infiltration and unfavorable outcomes[22,23]. As such, they are increasingly being recognized as important drivers of tumor progression with one mechanism being the formation of neutrophil extracellular traps (NETs), also termed NETosis. NETs, first discovered in 2004 as a mechanism of antimicrobial defense, are web-like structures extruded by PMNs composed of DNA and histones[24]. Recent studies suggest that exaggerated NET formation is implicated in the pathogenesis of autoimmune diseases[25,26], chronic inflammation[27], cancer-associated thrombosis[28], and adverse effects after surgical stress[29,30]. Further, NETs can capture circulating tumor cells and facilitate seeding to distal sites promoting metastasis[31]. Although the impact of NETs has been investigated in multiple aspects of tumor progression, it is important to highlight that their impact has not been explored in the context of cancer-related therapies, notably RT. RT plays an integral role in the clinical management of many solid tumors, and while NETs have been heavily implicated in inflammatory conditions, their role in radioresistance is unknown. This gap in our current understanding led us to investigate, whether NETs play a role in tumor radiation response.

## Results

### Radiation induces neutrophil extracellular trap formation.
Radiation-induced inflammation results in a transient and rapid

infiltration of PMNs to the TIME[13]. Although PMN infiltration can ameliorate radiation response, a recent study suggests that it may in fact promote tumor resistance to RT[21]. While it is unknown if this effect is NET-mediated, it is highly probable as NETs have been shown to drive PMN-facilitated tumor progression. We therefore sought to investigate if RT induces NET formation using a syngeneic heterotopic model of invasive urothelial carcinoma. MB49 cells were injected s.c into flanks of C57BL/6 mice and targeted radiation to the tumor was delivered using a low dose of 2 Gy, high dose of 10 Gy or two fractions of 5 Gy (Fig. 1a). Post-RT, tumors were collected for immunofluorescence analyses at an early timepoint (72 h of post-RT) and late timepoint (1-week of post-RT). Tissues were stained for PMNs (Ly6G) and NETs with neutrophil elastase (NE) and citrullinated histone-H3 (H3Cit), a NETs specific marker[32]. Compared to non-irradiated controls, increased PMN infiltration was noted in tumors that were radiated with a fractionated dose ($2 \times 5$ Gy) and high dose (10 Gy) 72 h of post-RT ($p < 0.05$ and $p < 0.001$) and 1-week of post-RT ($p < 0.001$); however no significant differences were observed when tumors were irradiated at a low dose (2 Gy) at either timepoint (72 h $p = 0.13$, 1 week $p = 0.11$) (Fig. 1b–d and Supplemental Fig. 1A). Moreover, irradiation significantly induced NET formation 72 h of post-RT in tumors that were irradiated with $2 \times 5$ Gy and 10 Gy ($p < 0.05$) and 1-week of post-RT all doses had a significant percentage increase in NETs compared to non-irradiated controls ($p < 0.001$). H&E staining revealed increased necrosis in tumors that were irradiated with single dose of 10 Gy 1-week post-RT compared to non-irradiated controls, and tumors radiated with 2 Gy or $2 \times 5$ Gy (Supplemental Fig. 1B), which may explain the increased infiltration of PMNs, and NETs observed in this group. To determine if the recruitment of PMNs and NET formation was dependent on the presence of the tumor, we examined PMN infiltration and NET formation in a (1) normal murine bladder, (2) bladder that was irradiated 10 Gy, (3) MB49 bladder tumor, and (4) MB49 bladder tumor that was irradiated 10 Gy (Supplemental Fig. 1C) Interestingly, in a normal murine bladder or irradiated murine bladder, no PMN infiltration or NET formation was observed. However, PMN infiltration was noted in tumor bearing bladders and irradiated tumor bearing bladders. Furthermore, NETs identified through NE and H3Cit staining were only observed in irradiated tumor bearing bladders, further demonstrating that irradiation of tumors promotes NET formation.

### NETs contribute to MIBC tumor radioresistance post-RT.
Since we noted an increased presence of NETs in the TIME of irradiated tumors compared to non-irradiated controls, we next sought to examine their role in tumor radiation response. To investigate this, MB49 cells were injected s.c into flanks of C57BL/6 or NETosis deficient mice. PAD4 is an essential enzyme in the NETosis pathway involved in the citrullination of histones-H3, a precursor to decondensation of chromatin in PMNs[33]. Moreover, it has been demonstrated that PAD4$^{-/-}$ mice are unable to form NETs in response to stimuli[34,35]. This was confirmed in our model through imaging flow cytometry as PAD4$^{-/-}$ tumors showed negative staining for H3Cit (Supplemental Fig. 2A) and was also confirmed through immunofluorescence staining for NE and H3Cit (Supplemental Fig. 2B). Post-tumor injection, mice were randomized and irradiated with two fractions of 5 Gy when tumors were palpable (0.2–0.3 cm$^3$). Mice were followed 2 weeks of post-RT for tumor growth analyses and till endpoint for survival (Fig. 2a). Tumor growth in the non-irradiated controls and NET deficient PAD4$^{-/-}$ showed no significant differences ($p = 0.28$); however irradiation of PAD4$^{-/-}$ mice significantly delayed tumor growth kinetics ($p < 0.001$) compared to irradiated controls

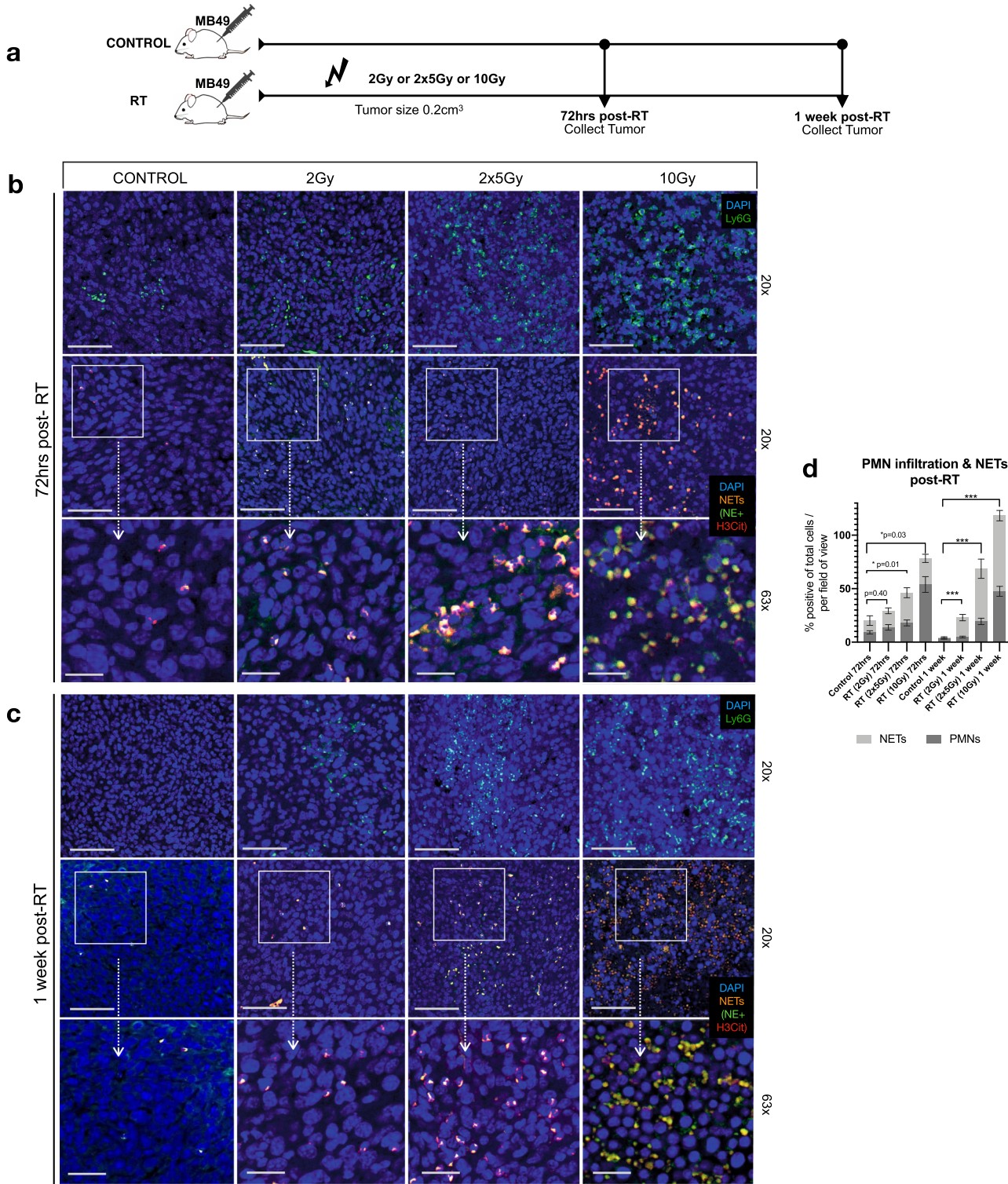

**Fig. 1 Radiation induces NET formation in murine MB49 tumors. a** Schematic representation of time point experiment where tumors were irradiated with: 2 Gy, two fractions of 5 Gy (2 × 5 Gy) or 10 Gy. Tumors were collected 72 h and 1-week post-radiation (post-RT) for immunofluorescence analyses. **b** Representative confocal images of non-irradiated controls or irradiated tumors 72 h post-RT or **c** 1 week-post RT stained for PMNs (Ly6G-green, 20×, 63×) and NETs, representative from three independent experiments (H3Cit-red and NE-green, 20×, 63×), nuclei (blue). Scale bars are 50 μm for 20× images and 20 μm for 63× images. **d** Quantification of PMNs and NETs in confocal images using QuPath V6, expressed as percentage of positive cells per total cells per field of view. Data are expressed as mean ± SEM ($n = 10$ mice per group, $n = 3$). Unpaired two-tailed students t-test was used to assess statistical significance between groups (***$p < 0.001$ for NETs).

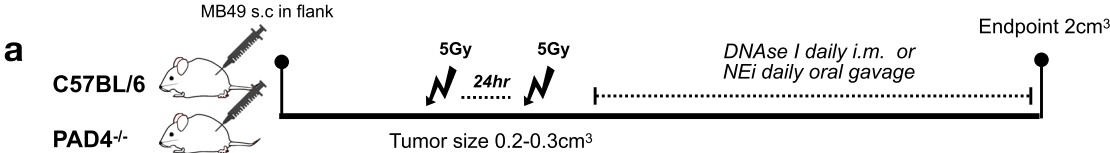

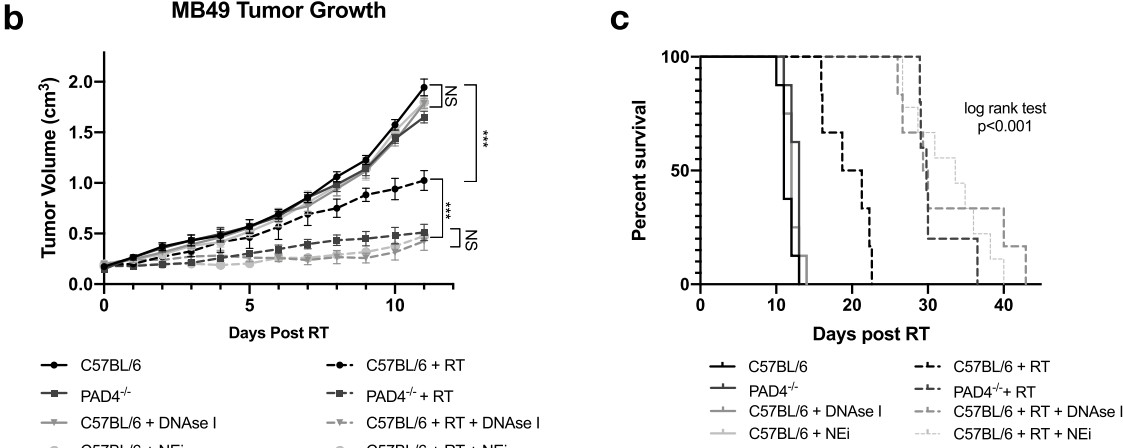

**Fig. 2 Inhibition and degradation of NETs improves radiation response. a** Schematic representation of tumor growth experiment, where C57BL/6 or PAD4$^{-/-}$ mice were injected subcutaneously (s.c) in the right flank with MB49 (500,000 cells). Mice were randomized into two groups: non-irradiated (control) or irradiated (RT). Tumors were irradiated with two fractions of 5 Gy and DNAse I was administered intramuscularly (i.m.) to degrade NETs or NEi was administered through oral gavage to inhibit NETs. **b** Tumor growth kinetics 11 days post-RT ($n = 8$ mice per arm). Data represented as mean ± SEM, two-way ANOVA with Bonferroni's multiple comparison's test was used to assess statistical significance. **c** Kaplan–Meier percent survival, log rank (Mantel-cox) test. NS = not significant $p < 0.05$ (C57BL/6 vs. PAD4$^{-/-}$ $p = 0.28$, C57BL/6 + DNAse I $p = 0.07$, C57BL/6 NEi $p = 0.10$, PAD4$^{-/-}$ RT vs. C57BL/6 RT + DNAse I $p = 0.34$, PAD4$^{-/-}$ RT vs. C57BL/6 RT + NEi $p = 0.06$, C57BL/6 RT + DNAse I vs. C57BL/6 RT + NEi $p > 0.99$), ***$p < 0.001$.

(Fig. 2b and Supplemental Fig. 2C). An increased overall survival was also noted (median survival: C57BL/6 + RT 20 d vs. PAD4$^{-/-}$ + RT 29 d, $p < 0.01$) (Fig. 2c). Moreover, in addition to utilizing the PAD4$^{-/-}$ model, we also evaluated other strategies to degrade/inhibit NETs. For instance, either by administering DNAse I, which degrades NET structures, or by inhibiting NET formation through administration of neutrophil elastase inhibitor (NEi). NE is expressed within NET structures and after translocation to the nucleus, it initiates NETosis by driving chromatin decondensation[36]. Decreased NET formation was confirmed in the tumors treated with DNAse I or NEi through immunofluorescence, where tumors showed decreased staining for H3Cit (Supplemental Fig. 2B). Similar findings were observed when mice were treated with DNAse I or NEi as compared to non-irradiated controls, no significant differences in tumor growth were noted ($p = 0.07$ or $p = 0.10$). However, compared to irradiated controls, a delay in tumor growth kinetics was observed in irradiated mice treated with DNAse I ($p < 0.001$) and improved overall response (median survival: 29 d, $p < 0.01$). Similarly, irradiated tumors treated with NEi also delayed tumor growth kinetics ($p < 0.001$) and improved radiation response compared to irradiated controls (median survival: 33 d vs. 20 d in irradiated controls, $p < 0.001$). Together these data demonstrate that inhibiting NETs through several NET-targeting approaches enhances the efficacy of RT.

**HMGB1 promotes NET formation through Toll-like-receptor 4.** Radiation in the TIME induces the release of several proinflammatory cytokines and damage-associated molecular pattern (DAMP) molecules, including the protein High Mobility Group

Box Protein-1 (HMGB1)[37]. HMGB1 is highly abundant in eukaryotes, where high expression has been reported in many tumor types including bladder cancer[38,39]. HMGB1 is a multifunctional protein which can be released by tumor cells post-RT, and triggers an array of DAMP-induced inflammatory responses[40,41]. Having shown that RT induces NETs, we next sought to gain a better understanding on what factor in the radiated TIME may be facilitating NET formation, and a suitable candidate was HMGB1. We have previously demonstrated that HMGB1 contributes to MIBC radioresistance through both its intracellular and extracellular functions[41,42]. To add, several studies report extracellular HMGB1 as potentiators of NET formation and contributors of PMN-associated inflammatory conditions[29,43]. Indeed, we like others, also observed that stimulation of human and murine PMNs with recombinant HMGB1 (rHMGB1) significantly induces NETs ($p < 0.001$) (Fig. 3a, b). Further, the addition of HMGB1 inhibitor glycyrrhizin (GLZ) prevents NET induction ($p < 0.001$). GLZ is a widely used natural inhibitor of HMGB1 as it physically binds to both HMG boxes and prevents its extracellular activity[44,45]. To examine if HMGB1 from irradiated tumor cells induces NET formation, conditioned media was collected from the human urothelial cancer cell line UM-UC3. Our previous findings demonstrated that high HMGB1 expression in this cell line contributes to MIBC radioresistance[42]. Human PMNs were isolated from blood of healthy donors and stimulated with UM-UC3 conditioned media. NETs were quantified through Sytox green fluorescence as Sytox is a cell impermeable dye that allows quantification of extracellular DNA (Fig. 3c, d). Stimulation of PMNs with irradiated UM-UC3 conditioned media significantly induced NET formation ($p < 0.001$), but this was not observed in

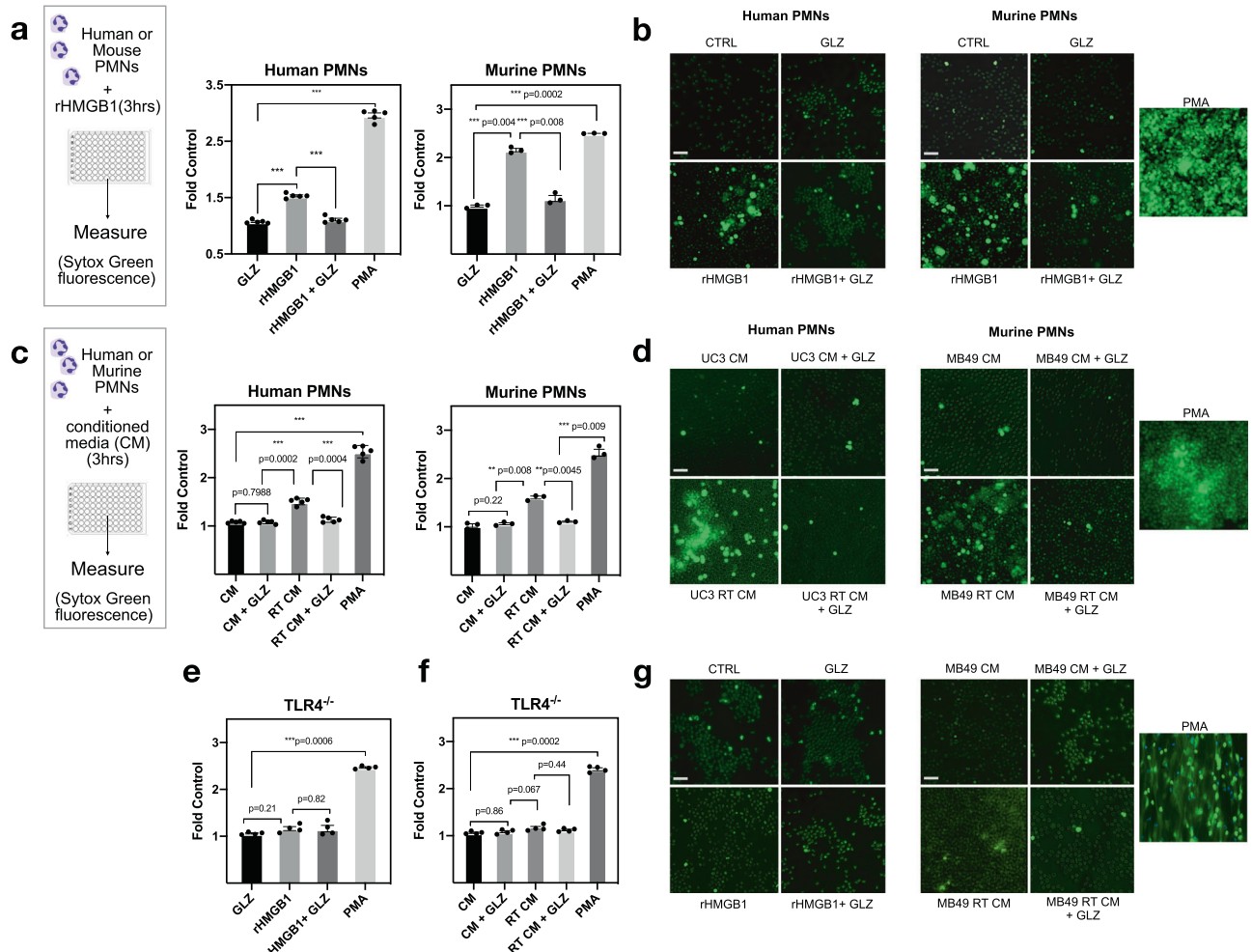

**Fig. 3 HMGB1 promotes NET formation through TLR4. a** Stimulation of human PMNs ($N = 5$ PMNs isolated from different healthy donors) or murine PMNs ($N = 3$) with rHMGB1 significantly induces NETs measured by Sytox green fluorescence ***$p < 0.001$. **b** Representative fluorescence images through confocal microscopy (10×), scale bar 100 μm. **c** Stimulation of human PMNs ($n = 5$ PMNs isolated from different healthy donors) with irradiated UM-UC3 conditioned media or murine PMNs ($n = 3$) with irradiated MB49 conditioned media significantly induces NETs and this is reversed through addition of GLZ. **d** Representative fluorescence images through confocal microscopy (10×), scale bar 100 μm. **e** Stimulation of murine PMNs from TLR4$^{-/-}$ mice stimulated with rHMGB1 ($n = 3$) or **f** irradiated MB49 conditioned media ($n = 3$), and **g** representative fluorescence images through confocal microscopy (10×), scale bar 100 μm. All data are represented as mean ± SEM of triplicates from individual experiments, paired-student's t-test was used to assess statistical significance (**a**, **c**, **e**, **f**).

the presence of GLZ ($p < 0.001$). Additionally, conditioned media was collected from MB49 cells where we have previously demonstrated that radiation increases the release of extracellular HMGB1[41]. The same was observed when bone marrow derived PMNs were isolated from C57BL/6 mice as, stimulation of murine PMNs with irradiated MB49 conditioned media significantly induced NET formation ($p < 0.01$); however this was reversed in the presence of GLZ ($p < 0.001$). To confirm that the source of extracellular DNA was from the PMNs and not cancer cells, we measured Sytox fluorescence after irradiation of UM-UC3 cells, MB49 cells or PMNs alone. No significant differences in Sytox fluorescence were observed when comparing irradiated tumor cells with tumor cells alone; however irradiation of PMNs led to a significant increase in fluorescence compared to non-irradiated controls (Supplemental Fig. 3). Toll-like receptor 4 (TLR4) is a known receptor of HMGB1 and several studies have demonstrated that HMGB1 induces NET formation in a TLR4 dependent manner and contributes to the severity of PMN associated inflammation[46,47]. To investigate if this was also the case in the context of radiation, we sought to investigate our

findings using PMNs from Toll-like receptor 4 knockout mice (TLR4$^{-/-}$). Indeed, stimulation of TLR4$^{-/-}$ PMNs with rHMGB1 did not induce NETs ($p = 0.21$); however, NETs were formed when stimulated with PMA (Fig. 3e). Similarly, stimulation of TLR4$^{-/-}$ PMNs with irradiated MB49 conditioned media did not induce NETs compared to non-irradiated controls ($p = 0.82$) (Fig. 3f, g). Together these findings highlight that HMGB1-dependent induction of NETs in the context of radiation is mediated through TLR4.

**Extracellular HMGB1 contributes to NET-mediated radioresistance.** Based on these in vitro findings and HMGB1's role in MIBC radioresistance, we hypothesized that inhibition of HMGB1 would improve radiation response by preventing HMGB1-mediated NET formation. To examine this, MB49 was injected s.c into flanks of mice and treated with two fractions of 5 Gy when tumors were palpable. Post-RT, GLZ was administered i.p. every alternate day to inhibit extracellular HMGB1 and DNAse I was administered daily i.m. to degrade NETs (Fig. 4a).

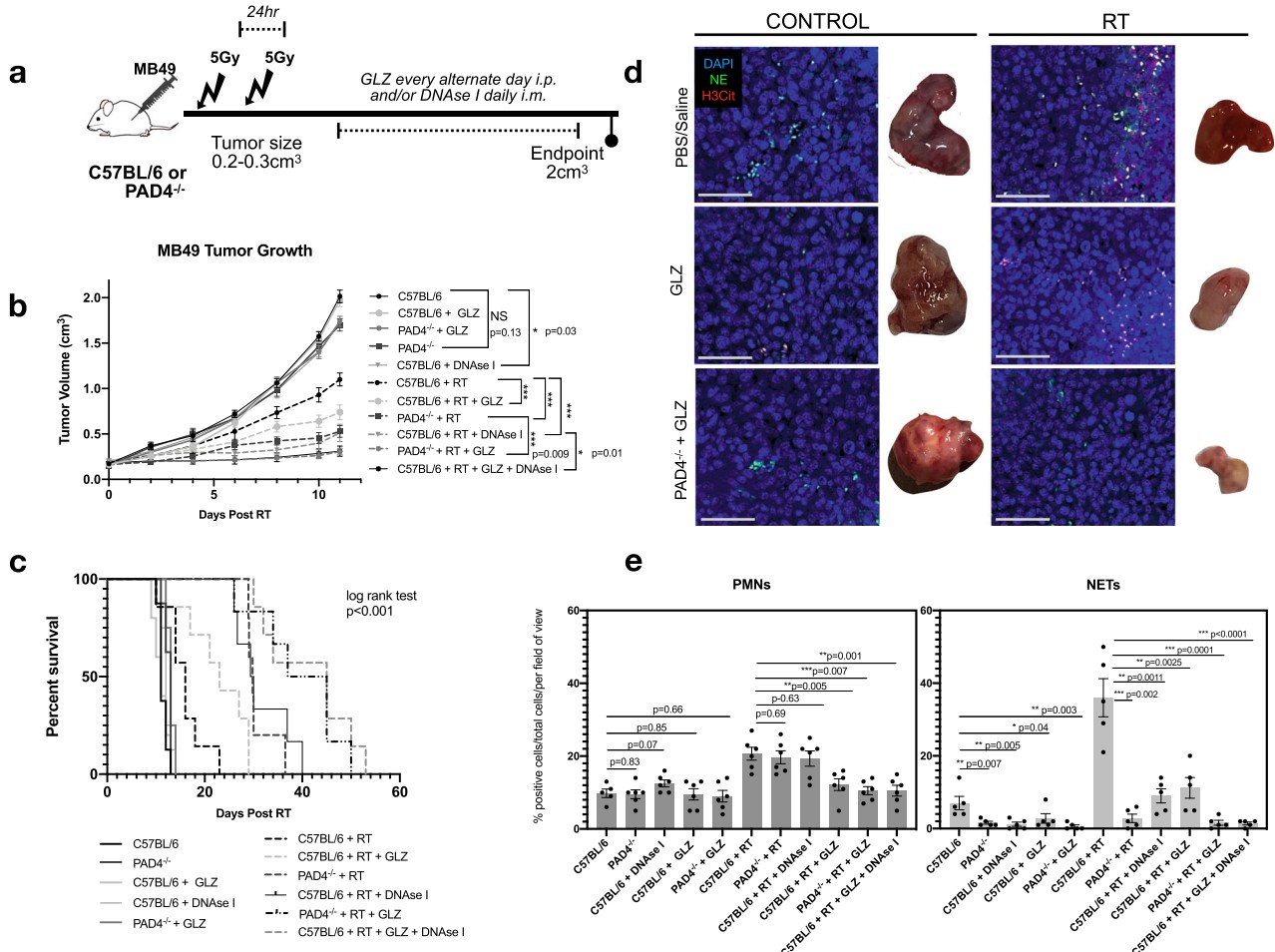

**Fig. 4 Inhibition of HMGB1 and NETs improves overall RT response. a** Schematic representation of tumor growth experiment where C57BL/6 or PAD4$^{-/-}$ mice were injected subcutaneously (s.c) in the right flank with MB49 (500,000 cells). Mice were randomized into two groups: non-irradiated (control) or irradiated (RT). Tumors were irradiated with two fractions of 5 Gy and DNAse I was administered intramuscularly (i.m.) to degrade NETs or GLZ was administered intraperitoneally (i.p) to inhibit extracellular HMGB1. **b** MB49 tumor growth 11 days post-RT ($n = 8$ mice per arm for all groups except $n = 10$ for C57BL/6 RT + DNAse I, $n = 9$ C57BL/6 + RT + GLZ + DNAse I). **c** Kaplan–Meier percent survival. **d** Immunofluorescence of tissues obtained at endpoint staining for PMNs and NETs through NE (green) and H3Cit (red) staining, nuclei (blue), 20×, scale bar 50 μm, representative of three independent experiments. **e** Quantification of PMN infiltration and NETs in tumors, data represented as percent positive of total cells per field of view ($n = 6$ mice). Data represented as mean ± SEM. Statistical significance was assessed using two-way ANOVA with Bonferroni's multiple comparison's test (**b**) log rank (Mantel-cox) test (**c**) unpaired two-tailed students *t*-test (**e**). NS not significant (compared to C57BL/6: C57BL/6 + GLZ $p > 0.05$, PAD4$^{-/-}$ + GLZ $p = 0.053$, PAD4$^{-/-}$ $p = 0.13$), ***$p < 0.001$.

Inhibition of HMGB1 post-RT significantly delayed tumor growth compared to irradiated controls ($p < 0.001$) and improved overall survival (median survival: C57BL/6 + RT + GLZ 23 d vs. 16 d in irradiated controls, $p < 0.05$) as previously reported[41]. Interestingly, inhibition of both HMGB1 and NETs significantly improved radiation response compared to irradiated controls treated with GLZ alone, DNAse I alone, or in PAD4$^{-/-}$ mice (C57BL/6 + RT + GLZ + DNAse I $p < 0.001$, PAD4$^{-/-}$ + RT + GLZ $p < 0.001$) (Fig. 4b and Supplemental Fig. 4). Furthermore, inhibition of both HMGB1 and NETs increased overall survival compared to irradiated mice treated with GLZ alone, DNAse I alone, or in PAD4$^{-/-}$ mice (median survival: PAD4$^{-/-}$ + RT + GLZ 41 d, C57BL/6 + RT + GLZ + DNAse I 45 d vs. C57BL/6 + RT + GLZ 23 d, C57BL/6 + RT + DNAse I 29 d, PAD4$^{-/-}$ + RT 29 d, $p < 0.001$) (Fig. 4c). Immunofluorescence analyses on these tissues at endpoint revealed decreased NET formation and PMN infiltration in PAD4$^{-/-}$ + GLZ mice ($p < 0.01$) and C57BL/6 + DNAse I + GLZ treated mice ($p < 0.01$) compared to irradiated controls (Fig. 4d, e). Furthermore, inhibition of GLZ and DNAse I decreased PMN infiltration in these tumors at endpoint

compared to RT + DNAse I alone ($p < 0.001$). Together, these data suggest that HMGB1 released in the TIME may promote NET formation, and inhibition of both NETs and HMGB1 can improve radiation response.

**DNAse I-treated mice show increased intratumoral CD8 infiltration post-RT.** NETs were first discovered as a mechanism of antimicrobial defense, serving as a physical trap to capture circulating fungi and bacteria[24]. Immunohistochemical analyses of NETs in inflammatory conditions reveal that they form a barrier between necrotic and viable tissues[48]. In the context of cancer, NETs can trap circulating tumor cells and have been hypothesized to facilitate escape from immune recognition, by forming a physical barrier between immune and tumor cells[31,49,50]. This prompted us to examine the spatial distribution of NETs and CD8 T-cells within the radiated TIME in our tumor model. To investigate this, we performed immunofluorescence on FFPE tumors collected at endpoint from the in vivo experiment outlined in Fig. 2 of: non irradiated controls (Control, DNAse I), irradiated mice (RT), and NET-degraded

irradiated mice (RT + DNAse I). Strikingly, confocal imaging of irradiated tumors demonstrated a similar phenomenon in our tumor model where, NETs formed a barrier between the tumor/stroma interface. NET clusters were observed at endpoint in the non-irradiated control tumors; however, the formation of a barrier was clearly evident in the irradiated tumors. Interestingly, CD8 T-cells were found behind this barrier and very few CD8 T-cell infiltration was noted inside tumor regions (Fig. 5a). On the contrary, irradiated NET degraded tumors (C57BL/6 + RT + DNAse I) showed notably increased intratumoral CD8 T-cell infiltration ($p < 0.001$) (Fig. 5b).

These observations led us to hypothesize that NETs may be contributing to tumor radioresistance by preventing intratumoral CD8 T-cell infiltration post-RT. To test this hypothesis, we examined MB49 tumor growth in athymic nude C57BL/6 mice as they lack CD4 and CD8 T-cells. Mice were irradiated as previously described, DNAse I was administered to degrade NETs and tumors were followed till endpoint (Fig. 5c). Interestingly, NET degradation following RT in C57BL/6 athymic mice did not improve response to radiation. No significant differences were observed in tumor growth kinetics (C57BL/5 + RT vs. C57BL/6 + RT + DNAse I, $p > 0.1$) nor survival ($p = 0.57$) (Fig. 5d, e). While athymic mice lack T-cells, they still contain other immune populations including PMNs. This was confirmed in our tumors through as PMNs were observed in all four arms (Fig. 5f, g). Moreover, irradiated tumors showed increased NET formation ($p < 0.01$) and NETs were degraded through treatment with DNAse I ($p < 0.001$).

**Evaluation of clinical relevance of NETs in human MIBC tumors.** Limited studies exist to date examining NETs in human tumors, and more importantly their implication in response to therapy. Elevated NETs have been noted in patients with lung, esophageal, and gastric cancers, where high levels of circulating NETs are predictive of advanced disease[51]. A study in 2013 histologically stained for NETs in a small cohort of patients with pediatric Ewing sarcoma, where tumoral NET deposition was associated with worse prognosis[49]. Another study reported markedly elevated levels of NETs in human liver and lung malignant tissues[52]. We sought to evaluate our preclinical findings in a cohort of human MIBC patients treated with RT-based therapy at our institution. Our aim was to investigate if NETs are present in the TIME of human MIBC tumors, and to examine the implication of PMN and CD8 T-cell infiltration in these tumors with outcome. To do so, we generated a TMA using a cohort of 104 MIBC patients treated with RT, where tissues were obtained from routine biopsy pre-RT and post-RT treatment. Post-RT treatment, patients were classified as RT non-responders evidenced by persistent viable cancer in post-treatment biopsies or as RT responders. Clinico-pathological characteristics of the patient population are outlined in Table 1. Chromogenic immunohistochemical staining was performed to examine infiltration of PMNs, CD8 T-cells, and NETs in this cohort (Fig. 6a). Our analysis revealed that NETs, stained with H3Cit and NE, were indeed present in the TIME of human MIBC patients, notably in post-RT treatment tissues. A significantly higher proportion of RT non-responders had NETs present in their TIME compared to RT responders (52% vs. 6%, $p < 0.001$) (Fig. 6b). Interestingly, further analysis confirmed NET deposition in the TIME post-RT was also associated with worse overall survival, even after adjusting for potential confounding factors (HR 3.75, 95% CI 1.67–8.44, $p < 0.001$—Supplemental Fig. 5 and Table 2). A sub-analysis on the NET positive proportion of RT non-responders was performed, to examine PMN infiltration pre-RT and post-RT. A significant trend was noted in these patients as they exhibited increased intratumoral PMN infiltration post-

RT treatment compared to their baseline pre-RT ($p < 0.01$) (Fig. 6c). This prompted us to investigate if pre-treatment intratumoral PMN infiltration correlates to post-treatment NETs. Indeed, among all RT non-responders, a significantly higher intratumoral PMN infiltration was noted in patients that had NET deposition in their TIME compared to the RT non-responders that were absent of NETs ($p < 0.05$) (Fig. 6d). This in main finding validates the notion that radiation-based therapy can induce NET deposition within the TIME of human bladder cancer patients, and these events are associated with response to RT. Moreover, the correlation of pre-treatment PMN tumor infiltration with NET formation suggest that it can be a predictive marker for likelihood of complete RT response.

A high PMN to CD8 ratio in the tumor has been shown to be a predictor of poor outcome in patients with resected esophageal and lung cancers[53,54]. This inverse correlation prompted us to examine if NETs impact the spatial distribution of CD8 T-cells in tumors as seen in our preclinical findings. Indeed, we observed a distinct spatial distribution of CD8 T-cells and NETs, where NETs were found to be surrounding CD8 T-cells but not colocalizing with them (Fig. 6e). In accordance with these findings, a higher intratumoral PMN to CD8 ratio was observed in our RT non-responders compared to RT responders ($p < 0.01$) (Fig. 6F). Further, a high PMN to CD8 ratio correlated with poor overall survival (HR 1.89, 95% CI 1.01–3.53, $p < 0.05$—Fig. 6G) and complete response (OR 0.21, 95% CI 0.05–0.75, $p < 0.05$—Table 2). When performing a multivariable analysis to adjust for possible confounders, PMN to CD8 ratio was still associated with complete response (OR 0.18, 95% CI 0.04–0.72, $p < 0.05$). Although this did not independently associate with worse overall survival, these data need to be interpreted with caution as a limitation of this analysis is a relatively small sample size. As such, we decided to look at the PMN to lymphocyte ratio (NLR) in the blood as a high NLR has been shown to correlate with poor prognosis in patients with various tumor types[19]. Using a larger cohort of patients that have been treated with RT-based therapy at our institution, we observed that a high NLR in the blood was independently associated with poor prognosis marked by worse overall survival ($p < 0.01$) (Supplemental Fig. 6) and this was true when accounting for possible confounders through a multi-variable analysis (HR 1.75, 95% CI 1.13–2.73, $p < 0.05$). Together these observations in tumors of patients with MIBC support our preclinical findings and suggesting presence of PMNs and NETs can impact response to RT.

## Discussion

In this study we provide evidence demonstrating a critical role for NETs in tumor radioresistance. We show that irradiation of urothelial cancer cells correlates with NET formation in vitro, in vivo, and in humans with MIBC. Furthermore, irradiation of NETosis deficient PAD4$^{-/-}$ mice or C57BL/6 mice treated with NET degradation/inhibition significantly improves overall radiation response. This study demonstrates a correlation between the NET inducing effect of radiation on a human cancer, namely in human MIBC tumors treated with RT-based therapy. Specifically, these were observed in post-treatment tumors of patients with persistent disease. In this study, we explored upstream how the radiated TIME promotes NET formation by highlighting the role of HMGB1. Further investigations exploring the downstream effect of NETs in radioresistance and their interplay with other components of the TIME are warranted, such that a therapeutic window of opportunity could be identified.

RT is known to induce immunogenic cell death, through the release of DAMP's which promotes anti-cancer CD8 T-cell responses[55]. However, the release of DAMP's in the TIME such

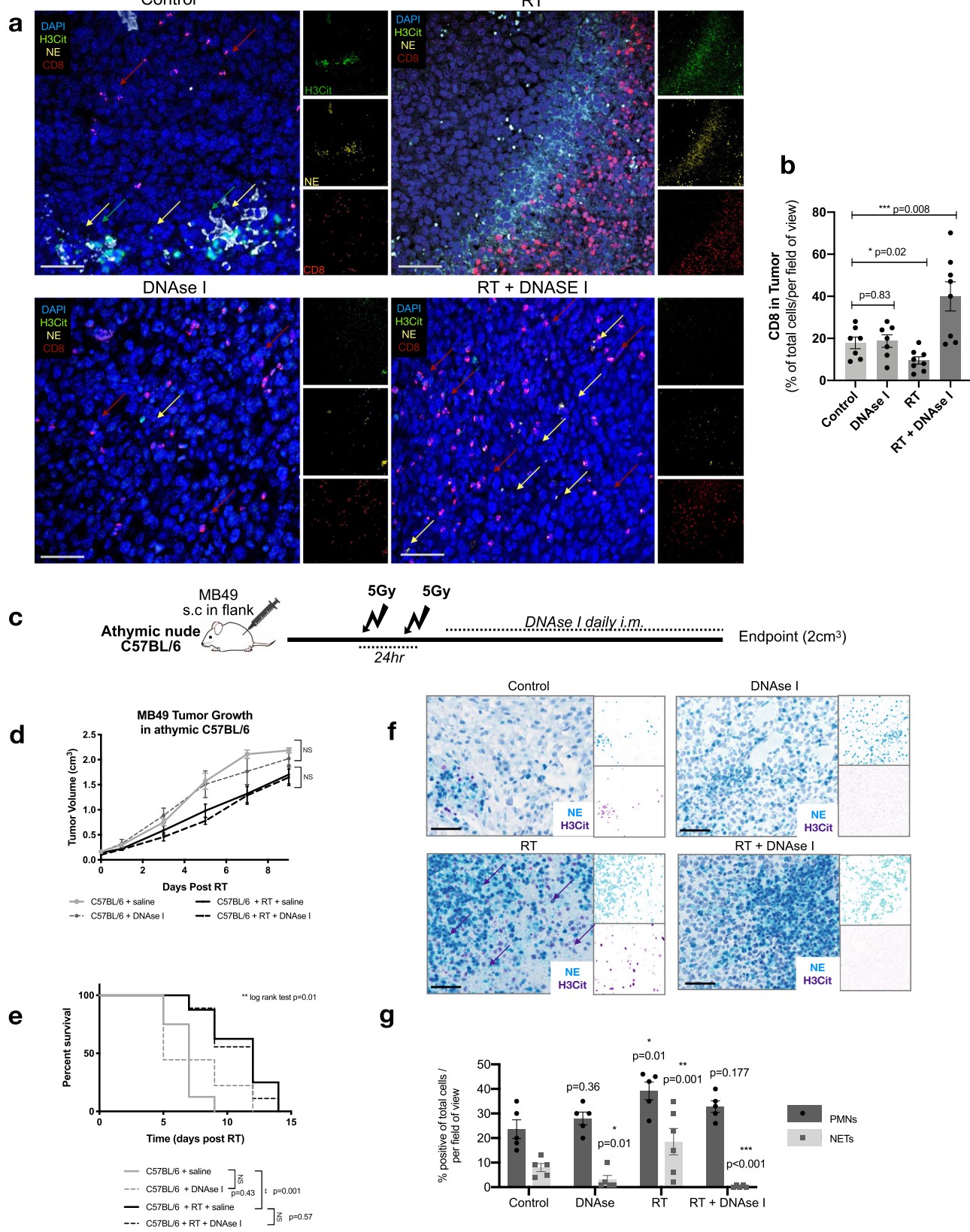

as HMGB1 has also been shown to promote immunosuppressive pathways resulting in tumor immune escape contributing to tumor radioresistance[56]. Indeed, we have previously demonstrated that the release of extracellular HMGB1 in TIME post-RT contributes to MIBC radioresistance by shifting the TIME towards a tumor promoting landscape[41]. Inhibiting HMGB1 in irradiated MB49 tumors significantly impairs infiltration of myeloid derived suppressor cells (MDSC) and tumor associated macrophages (TAMs) in the TIME, and shifts the overall tumor immune balance towards an anti-tumoral response. As such to gain a better understanding

**Fig. 5 DNAse I-treated mice show increased intratumoral CD8 infiltration post-RT. a** FFPE tumors were collected from C57BL/6 mice injected subcutaneously (s.c) with MB49 tumors from groups: non-irradiated control (Control), C57BL/6 mice treated with DNAse-treated mice (DNAse I), irradiated C57BL/6 (RT) and irradiated C57BL/6 treated with DNAse I (RT + DNAse I) at endpoint (3 weeks post-RT). Tissues were stained for immunofluorescence analyses with CD8 (red), NE (yellow), H3Cit (green), nuclei (blue). Representative confocal images from three independent experiments obtained with 20× objective, scale bar 50 μm. **b** Quantification of intratumoral CD8 T-cell infiltration using QuPathV6. Data represented as percentage CD8 T-cells per total cells per field ($n = 7$ mice per arm). **c** Schematic of tumor and survival experiment in athymic C57BL/6 mice. **d** Tumor growth kinetics of irradiated athymic mice injected s.c with MB49, treated with ($n = 8$ mice per group) or without DNAse I ($n = 9$ mice per group). **e** Kaplan–Meier percent survival. **f** Chromogenic immunohistochemical staining of athymic tumors, stained for PMNs and NETs (NE-blue, H3Cit-purple), scale bar 50 μm. **g** Quantification of PMN infiltration and NETs in tumors, data represented as percent positive of total cells per field of view ($n = 5$ mice per arm). Data represented as mean ± SEM, unpaired two tailed student's $t$-test was used to assess statistical significance in (**b**, **g**) two way ANOVA with Bonferroni's multiple comparison's test (**d**), log rank (Mantel-cox) test (**e**). NS not significant ($p > 0.05$), *$p < 0.05$, **$p < 0.01$, ***$p < 0.001$.

**Table 1 Clinico-pathological characteristics of patient cohort in TMA analysis. Chi-squared test was used to assess statistical significance.**

| | Responders $n = 71$ (68.27%) | | Non-responders $n = 33$ (31.73%) | | Total $n = 104$ | | |
|---|---|---|---|---|---|---|---|
| | $n$ or median | % or IQR | $n$ or median | % or IQR | $n$ or median | % or IQR | $p$-value |
| Age | 75 | 68–80 | 75 | 64–79 | 75 | 65–80 | 0.5255 |
| Gender | | | | | | | |
| Male | 54 | 77.14 | 25 | 75.76 | 79 | 75.96 | 0.8767 |
| Female | 16 | 22.86 | 8 | 24.24 | 24 | 23.08 | |
| ECOG | | | | | | | |
| 0 | 32 | 45.71 | 15 | 45.45 | 47 | 45.2 | 0.3789 |
| 1 | 20 | 28.57 | 13 | 39.39 | 33 | 31.73 | |
| 2 | 17 | 24.29 | 2 | 6.06 | 19 | 18.27 | |
| 3 | 1 | 1.43 | 3 | 9.09 | 4 | 3.85 | |
| Tumor stage | | | | | | | |
| T2 | 66 | 95.31 | 27 | 96.3 | 93 | 83.65 | 0.791 |
| T3 | 5 | 3.13 | 6 | 3.7 | 11 | 2.88 | |
| CIS | | | | | | | |
| No | 37 | 52.11 | 17 | 51.52 | 54 | 51.92 | 0.8588 |
| Yes | 26 | 36.62 | 11 | 33.33 | 37 | 35.58 | |
| LVI | | | | | | | |
| Absent | 40 | 57.14 | 17 | 51.52 | 57 | 54.81 | 0.3386 |
| Present | 18 | 25.71 | 12 | 36.36 | 30 | 28.85 | |

of how the irradiated TIME facilitates and potentiates NET formation, we highlight the role of HMGB1 where we observed that stimulation of human and murine PMNs with rHMGB1 or irradiated conditioned media directly induces NETs. However, in the presence of HMGB1 inhibitor GLZ, no significant effect was observed. Importantly, our work confirms earlier work done by Tadie et al. and Tohme et al., which demonstrated a role for HMGB1 in promoting NET formation in the contexts of chronic inflammation and ischemia reperfusion injury[29,46]. Tadie et al. observed that HMGB1 induced NET formation was independent of NADPH oxidase reactive oxygen species (ROS) production, and highlight that it functions through a TLR4 dependent manner. We confirm this work in vitro by demonstrating in the context of radiation, HMGB1 induces NET formation in a TLR4 dependent manner as stimulation of TLR4$^{-/-}$ PMNs with rHMGB1 or irradiated conditioned media had no effect. Moreover, in vivo, inhibition of HMGB1 and NETs further improved radiation response. These findings provide key insights for HMGB1's role in the TIME post-RT, and how it contributes to tumor radioresistance.

While we highlight one mechanism through which RT facilitates NET formation in the TIME, we accept that HMGB1 may not be the sole factor. RT induces several dynamic changes in the TIME, and an important component is tumor hypoxia. Tumor hypoxia is a common feature in solid tumors and since RT increases levels of ROS in the TIME, hypoxia-mediated

radioresistance can be a challenge[57]. Hypoxia-inducible factor-1α (HIF-1α) is stabilized under hypoxic conditions, and plays a role in the upregulation of cell survival genes. Moreover, genetic and pharmacological inhibition of HIF-1α has been shown to decrease NET formation[58]. This can be explained by the two pathways that PMNs can undergo NET formation: late suicidal NETosis (NADPH oxidase dependent), or early vital NETosis which is NADPH independent[59]. Though we did not explore this in our study, the impact of RT induced ROS on the generation of NETs is an area that can be explored further.

Immunofluorescence analyses on irradiated MB49 tumors treated with or without DNAse I revealed a distinct spatial distribution between NETs and CD8 T-cells. A barrier of NETs was observed in irradiated tumors between the tumor and stroma with CD8 T-cells located behind this barrier. Meanwhile, increased intratumoral CD8 T-cell infiltration was noted in tumors of mice treated with DNAse I. Chromogenic staining for NETs in human MIBC tumors revealed a similar finding where a barrier of NETs was found surrounding CD8 T-cells near the tumor/stroma interface. Our observations support earlier findings by Bilyy et al., who demonstrated that patients with acute abdominal inflammation form a barrier of NETs between necrotic and viable tissue[48]. This was also noted by Berger-Achituv et al., whose study was the first study to show NETs in tumor beds of patients with Ewing sarcoma[49]. Staining for NETs in biopsy samples revealed that they were spatially located

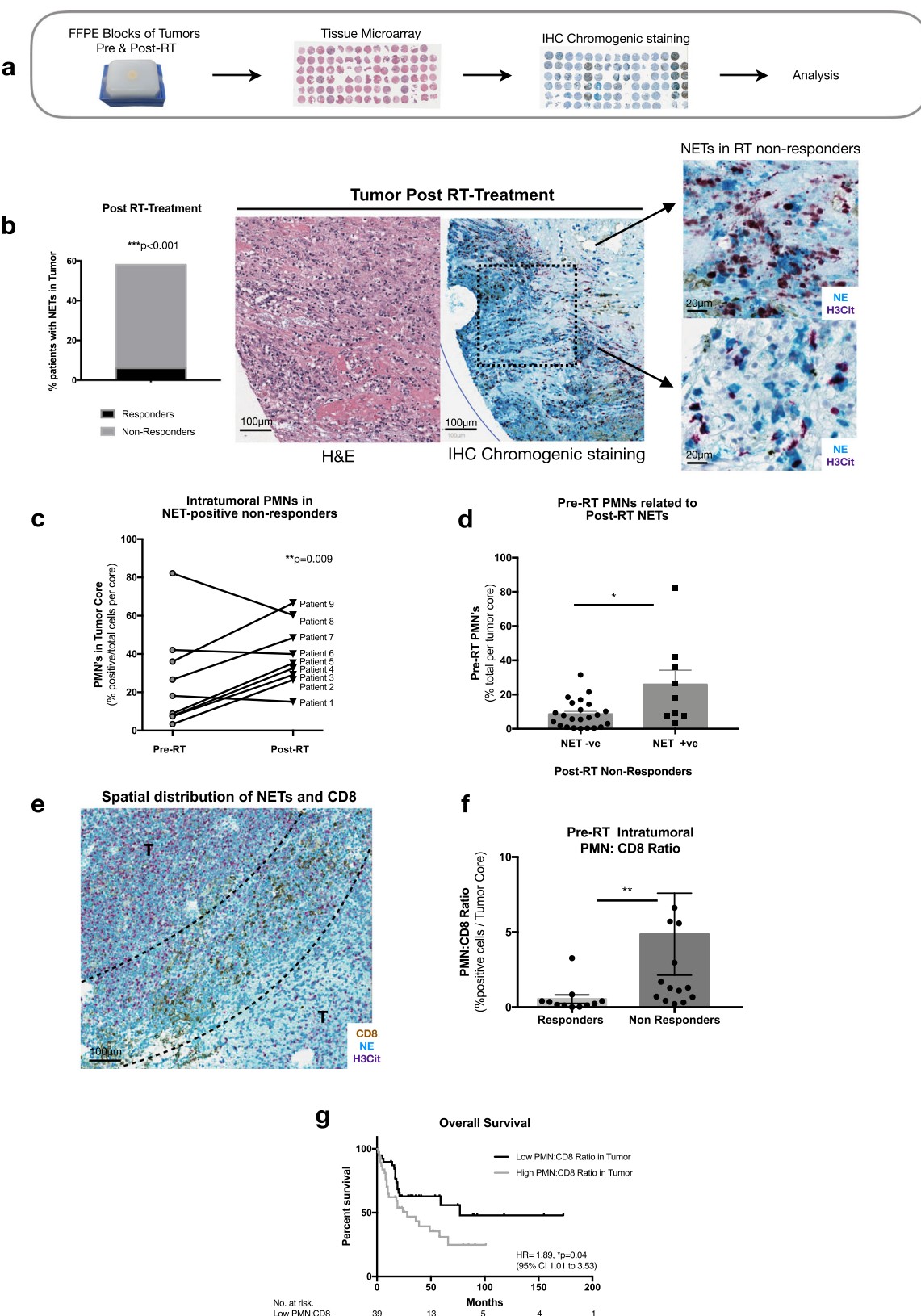

between the interface of tumor cells and necrotic tissue, and the authors proposed that this NET-barrier may be facilitating immune escape. This in fact was recently demonstrated by Teijeira et al., where they show through intravital microscopy that intratumoral NETs physically obstruct contact of tumor cells from CD8 T-cell and NK cells in the TIME[60]. This is the first study to demonstrate a direct link between intratumoral NETs and immune cytotoxicity. Our in vivo results support these findings as inhibition of NETs in athymic C57BL/6 mice did not improve radiation response compared to irradiated controls. These findings are in line with a previous report where PMNs were shown to play an immunosuppressive role in the TIME.

**Fig. 6 NETs are present in human MIBC tumors and a high PMN to CD8 ratio is associated with worse overall survival. a** Schematic representation of retrospective study design. **b** Representative image of NETs staining in MIBC tumors, where NETs were observed in patients with persistent disease post-RT treatment compared to responders, two-sided Fischer's exact test ($n = 52\%$ non responders, $n = 6\%$ responders). **c** Intratumoral PMN infiltration pre-RT and post-RT treatment in subset of patients whom were NET positive, unpaired two-tailed paired student $t$-test. **d** Pre-RT intratumoral PMN's correlates with post-RT NETs, unpaired two-tailed Mann–Whitney test ($p = 0.02$) (NET $+$ ve patients $n = 9$, NET$-$ve patients $n = 23$). **e** Representative image of the spatial distribution of NETs (H3Cit-purple, NE-blue) and CD8 (brown) observed in tumor of a non-responder, tumor region (T). **f** PMN to CD8 ratio pre-RT, data expressed as percentage of positive cells per tumor core, unpaired two-tailed Mann–Whitney test $p = 0.0018$ ($n = 11$ responders, $n = 14$ non-responders). **g** Overall survival of patients with low or high PMN to CD8 ratio, log rank test (Mantel-cox). Data represented as mean ± SEM, *$p < 0.05$, **$p < 0.01$, ***$p < 0.001$.

**Table 2 Univariate and multivariable analysis on patient cohort in TMA analysis.**

| | Univariate analysis | | | |
| | Logistic regression—CR | | Cox regression—OS | |
| | OR (95% CI) | *p*-value | HR (95% CI) | *p*-value |
|---|---|---|---|---|
| Age (year) | 1.01 (0.96–1.06) | 0.71 | 1.03 (1.00–1.06) | 0.05 |
| Gender (Male vs. female) | 0.48 (0.17–1.37) | 0.17 | 1.05 (0.57–1.95) | 0.88 |
| BMI pre-treatment | 1.05 (0.96–1.15) | 0.27 | 0.94 (0.89–0.99) | 0.03 |
| ECOG (0/1 vs. 2/3) | 1.64 (0.33–8.23) | 0.55 | 3.3 (1.90–5.73) | <0.001 |
| Smoking (Never vs. past/current) | 0.33 (0.10–1.08) | 0.07 | 1.47 (0.82–2.64) | 0.2 |
| Diabetes (No vs. Yes) | 0.67 (0.24–1.87) | 0.44 | 1.04 (0.59–1.84) | 0.89 |
| LVI (No vs. Yes) | 0.51 (0.19–1.41) | 0.20 | 1.81 (1.05–3.12) | 0.03 |
| Tumor Stage (T2 vs. T3) | 0.66 (0.15–2.88) | 0.58 | 3.36 (1.74–6.47) | <0.001 |
| PMN/CD8 tumor (Low vs. High) | 0.21 (0.05–0.75) | 0.02 | 1.89 (1.00–3.53) | 0.04 |
| NETs post-RT (No vs. Yes) | NA | | 2.64 (1.23–5.62) | 0.01 |

| | Multivariable analysis | | | |
| | Logistic regression—CR | | Cox regression—OS | |
| | OR (95% CI) | *p*-value | HR (95% CI) | *p*-value |
|---|---|---|---|---|
| LVI (No vs. Yes) | | | 2.22 (1.15–4.28) | 0.02 |
| PMN/CD8 in tumor (Low vs. High) | 0.18 (0.04–0.72) | 0.02 | | |
| NETs post-RT (No vs. Yes) | NA | | 3.75 (1.67–8.44) | <0.001 |

Multivariable logistic regression was performed to assess the role of each covariates on a complete response post-RT. In order to estimate the impact of each variable on overall survival a Cox model was performed. Log linearity hypothesis were verified for quantitative variables and proportional hazard assumptions were verified for all variables. Final multivariate model was created using a stepwise selection method. Statistical significance was set at 5%.
*CR* complete response, *OS* overall survival, *OR* odds ratio, *HR* hazard ratio, *BMI* body-mass index, *NA* Non applicable.

CXCR2 inhibition of PMNs in pancreatic tumors results in increased tumoral T-cell infiltration and depletion of PMNs in tumor bearing mice increases CD8 T-cell activity resulting in an anti-tumor response[17,61]. Retrospective analyses in our cohort revealed that patients with persistent disease had a high pre-treatment intratumoral PMN to CD8 ratio which correlated to worse overall survival. Together, these findings support an interplay between PMNs, NETs, and CD8 T-cells which contribute to MIBC tumor radioresistance, and provide a foundation for prospective investigations. Future studies examining the spatial immune distribution will provide valuable information on other immunological players that may be involved. Further, functional studies examining the role of NETs within the TIME are warranted to gain a deeper understanding of the mechanisms behind NET induced radioresistance in the TIME. Although the findings from the human tumors complement our preclinical work, we acknowledge that the study has its limitations. Due to the retrospective nature of our study, factors such as small tumor core required to generate the TMA, small sample size in terms of responders and non-responders are limitations that may underestimate the role of NETs in our cohort. Nonetheless, our findings provide evidence of NETs in the TIME of human MIBC tumors.

In summary, we demonstrate a role for NETs as promoters of radioresistance in a heterotopic model of invasive bladder cancer. Therapeutic intervention targeting NETs such as DNAse I explored in this study could be expanded in clinical settings, as they are currently used clinically and show no toxicity[62]. These findings provide key insights about the extrinsic influence of the TIME post-RT and can help increase the efficacy of RT for the management of MIBC and other cancers.

## Methods

**Cell lines and cell culture**. Human urothelial carcinoma cells UM-UC3 (Sigma-Aldrich, Oakville, ON, Canada) were cultured in Eagle's Minimum Essential Medium (EMEM, Wisent, St-Bruno, QC, Canada). Murine bladder cancer cells MB49 were cultured in Dulbecco's Modified Eagle Medium (DMEM, Wisent) and were a gift from Dr. Peter Black (University of British Columbia, Canada). MB49 cells originate from carcinogen-induced bladder carcinoma in C57BL/6 mice, and share similarities with human bladder cancer in terms of surface markers and tumor immune profile[63,64]. All cell lines were supplemented with 10% fetal bovine serum (FBS, Wisent) and incubated at 37 °C, 5% $CO_2$. Cells were routinely passaged at 70% confluency.

**In vitro irradiation**. UM-UC3 cells or MB49 cells were seeded in 100 mm plates and were irradiated when they reached 60–70% confluency. Prior to irradiation, cells were supplemented with fresh media and irradiated 6 Gy using a Faxitron X-Ray machine (Faxitron, Tucson, AZ, USA). Conditioned media was collected 24 h of post-RT and media was collected from non-irradiated controls. Conditioned media was concentrated using Amicon Ultra 10 K tubes (Cat #: UFC901008, Millipore Sigma, Burlington, MA, USA).

**In vitro quantification of NETs**. Human PMNs were isolated from healthy subjects using Ficoll Hypaque density centrifugation[65]. Mouse bone marrow derived

PMNs were isolated from bone marrow of tibias and femurs using a Histopaque (1077/1119) gradient[66]. Only isolates with >98% purity and viability as determined by Turk's staining and Trypan Blue exclusion were used. Fresh PMNs ($4 \times 10^5$) were seeded in CoStar 96-well plates supplemented with 3% RPMI (Wisent) and were incubated for 3 h with: phorbol 12-myristate 13-acetate (PMA, Cat#: P1585, 100 nM, Sigma-Aldrich), 3% RPMI, Glycyrrhizin (GLZ, Cat#: CDSO20796, 200 μM, Sigma-Aldrich), recombinant HMGB1 (rHMGB1, Cat#: 557804, 50 ng, Biolegend, San Diego, CA, USA), rHMGB1 and GLZ (50 ng, 200 μM), UM-UC3 or MB49 conditioned media, irradiated UM-UC3 or MB49 conditioned media. Sytox Green (Cat#: S7020, 5 μM, Life Technologies, Burlington, ON, Canada) was added 30 min prior to imaging and fluorescence was measured using a microplate reader at an excitation wavelength of 485 nm and emission of 535 nm. Representative well images were captured using an inverted fluorescent microscope with the (Nikon Eclipse TE300, 10×/0.25 Ph1 DL lens).

**Animals.** Seven to ten weeks old male C57BL/6 mice and athymic C57BL/6 mice were obtained from Charles River Laboratories (Senneville, QC, Canada). Peptidyl arginine deiminase type IV knockout (PAD4$^{-/-}$) mice were a gift from Dr. Allan Tsung (The Ohio State University, USA). Tlr4$^{tm1.2Karp}$ (TLR4$^{-/-}$) mice were obtained from the Jackson Laboratory (Stock#029015, Bar Harbor, ME, USA). All animals were maintained in a pathogen free-environment, at a temperature of 21 °C +/− 1 °C, 40–60% RH +/− 5% RH humidity, light-dark cycle altered every 12 h at the animal facility of the Research Institute McGill University Health Center (RI-MUHC). All animal experiments were performed according to the Canadian Council on Animal Care All procedures were approved by the McGill University Animal Care Committee (Protocol #7585) at our facility.

**In vivo tumor growth model.** A syngeneic model was used where MB49 (500,0000 cells) were subcutaneously (s.c) implanted into the right flank of C57BL/6 or PAD4$^{-/-}$ mice. When tumors reached a volume of 0.2–0.3 cm$^3$, mice were randomized and irradiated using the X-RAD SmART Irradiator Pxi 225cx (Precision X-ray, North-Branford, CT, USA). Mice were anaesthetized and two fractions of 5 Gy of radiation were delivered 24 h apart to the subcutaneous tumor to give a total dose of 10 Gy using the settings 225 kV, 13 mA, 0.3 mm copper filter. Tumor growth was monitored till endpoint (2 cm$^3$) using an electronic digital caliper and tumor volumes were calculated using an ellipsoidal approximation formula $V = [(\text{length} \times \text{width}^2) \times (\pi/6)]$. HMGB1 was modulated through intraperitoneal (i.p.) injections of GLZ (50 mg/kg, Sigma-Aldrich) and was administered 4 h post-RT, and every alternate day till endpoint. NETs were modulated through intramuscular injections (i.m.) of DNAse I (Cat#: 04536282001, 2.5 mg/kg, Roche Diagnostics, Laval, QC, Canada) or gavage of neutrophil elastase inhibitor (NEi) (Cat#: GW311616A, 2.2 mg/kg, Abcam, Cambridge, MA, USA). DNAse I or NEi was administered 24 h prior RT and then daily till endpoint. As controls, mice were given the vehicle treatment (PBS or saline alone).

**Immunofluorescence.** Tumors were formalin fixed and paraffin embedded (FFPE) and sectioned (5 μm) by the histopathology platform at the RI-MUHC. All immunofluorescence steps were carried out according to the Cell Signaling Technology immunofluorescence protocol. Antigen retrieval was performed using Sodium Citrate buffer pH 6 for all slides. Primary antibodies: H3Cit (Cat#: ab5103, 1:100, Abcam), Ly6G (Cat#: 551459, 1:100, BD Biosciences, San Jose, CA, USA), CD8 (Cat#: 98941, 1:100, Cell-Signaling Technologies, Danvers, MA, USA), NE (Cat#: Bs-698-2R, 1:100, Bioss USA, Woburn, MA, USA), DAPI (Cat#: D9542, 0.5ug/ml, Sigma-Aldrich). Secondary antibodies: AF488 (Cat#: A-11006, 1:1000, Thermo Fisher Scientific, Waltham, MA, USA), NE (Cat#: Bs-698-2R-AF488, 1:100, Bioss USA), H3Cit-AF568 (ab5103, was conjugated to AF568 Cat#: A-20184, Thermo Fisher Scientific). Multiplex-IHC was performed using the kit from Perkin Elmer (Cat#: NEL810001KT, Waltham, MA, USA) using fluorophores Opal 520, Opal 570, Opal 620. Slides were visualized using a Zeiss LSM 780 laser scanning confocal microscope using the 20×/0.4 LD Plan-Neofluar, 40×/0.60 oil corr LD "Plan-Neofluar" and 63×/1.40 DIC Plan-Apochromat lenses (Zeiss, Dorval, QC, Canada) and the Zeiss LSM 880 ElyraPS1 laser scanning confocal microscope using the 20×/0.8 Plan-Apochromat lens at the imaging platform of the RI-MUHC.

**Quantification of PMN and NETs in issues.** Tumor regions were identified through hematoxylin and eosin (H&E) staining and these regions were used to obtain the immunofluorescence images through confocal microscopy. During analysis, tumor regions were confirmed again through DAPI staining by examining histological features such as nucleus size. Quantification was performed using the positive cell detection algorithm on QuPath version 6[67] and all detections were manually verified. PMNs were identified through positive staining for Ly6G or NE along with DAPI and NETs were identified through co-staining for NE and H3Cit.

**Imaging flow cytometry.** Tumors were extracted for tumor digestion using the Tumor Dissociation Kit (Cat#: 130-096-730, Miltenyi Biotec, Somerville, MA, USA) and GentleMACS Dissociator (Miltenyi Biotec). Single suspensions were strained through 70 μm filters and ACK lysis buffer (Cat#: A1049201, Thermo Fisher Scientific) was used to remove red blood cells. $2 \times 10^6$ cells were stained with DAPI (1:1000, Sigma-Aldrich), Ly6G-FITC (clone 1A8, 1:200, Thermo Fisher Scientific) and H3Cit-AF647 (Cat#: ab5103, 1:100, Abcam).

**Retrospective cohort analyses.** Our retrospective cohort was composed of 104 MIBC patients that were treated with trimodal therapy, which includes radical transurethral resection of the bladder tumor (TURBT) followed by radiation-based therapy at the MUHC from 2004 to 2018. Pre-treatment tumor biopsies and routine bladder biopsy samples 1-month post-RT were collected, stored, and used under IRB approval (REB- RI-MUHC #2017-2612) and informed consent was obtained for all patients. The main clinical and histopathological characteristics of the patient population are summarized in Table 1. FFPE blocks were stained with H&E and tumor, stroma, and benign regions were identified and reviewed by a pathologist. Pre-treatment tissue microarray (TMA) were generated by sampling five cores (1.5 μm-thick) per patient (two tumor cores, two tumor-stroma transition cores, and one benign tissue core). For the post-treatment TMA, one core of tumor or previous tumor site tissue was utilized.

**TMA analysis.** Chromogenic immunohistochemical staining was performed using the Discovery Ultra Automated Immunostainer (Ventana Medical Systems, Tucson, AZ, USA) by the RI-MUHC Histopathology platform. Pre-treatment and post-treatment TMAs were stained for NETs and PMNs with H3Cit (Cat#: ab5103, 1:100, Abcam), NE (Clone: 950334, 1:100, Novus Biologicals, Oakville, ON, Canada) and CD8 (Clone: SP57, 1:100, Roche Diagnostics). TMAs were scanned using the Aperio Slide Scanner and slides were analyzed using QuPath software V6. TMAs were analyzed following the script created to analyze CD3 using fast cell counts[67].

**Statistical analyses.** GraphPad prism software version 9 was used for statistical analyses. All data are presented as mean ± SEM. Differences between groups were performed using the two-tailed student's $t$-test, Mann–Whitney test or one way ANOVA with Tukey's post-hoc. Tumor growth data were analyzed using two-way analysis of variance (ANOVA) with Bonferroni post-hoc test. Differences in survival curves were compared using the log-rank (Mantel-Cox) test. Multivariable logistic regression was performed to assess the role of each covariates on a complete response post-RT. A cox model was performed to estimate the impact of each variable on overall survival. Final multivariate model was created using a stepwise selection and was performed using SAS software (SAS Institute version 9.2, Cary, NC, USA). A $p$-value <0.05 was considered as statistically significant.

**Reporting summary.** Further information on research design is available in the Nature Research Reporting Summary linked to this article.

## Data availability

The authors declare that the data supporting the findings of this study are available within the paper and its supplementary information files. Source data are provided with this paper.

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

## Author contributions

S.S.J., J.J.M., R.R., J.S., and W.K. conceived and designed the experiments; S.S.J., M.A., R.S., R.K., F.Bo, G.M., and F.Br contributed to data acquisition; S.S.J., J.J.M., R.R., J.S., and W.K. analyzed and interpreted the data; S.S.J., J.J.M., R.R., J.S., and W.K. wrote and edited the manuscript.

## Competing interests

The authors declare no competing interests
