## [Peer Review File · Nature Communications]

Reviewers' Comments:

Reviewer #1:

Remarks to the Author:

The authors are investigating an important question – immune-related mechanisms of resistance to radiation therapy in bladder cancer. They have elegantly demonstrated that neutrophil extracellular traps contribute to resistance, and suggest that they may accomplish this by building a barrier between infiltrating CD8+ T-cells and the tumor. They have used subcutaneous mouse models of bladder cancer, and include some correlative human patient data to confirm the clinical relevance.

The paper is nicely written and is easy to read. The experiments are adequately described and the results are clearly presented. The authors do a particularly good job of “telling the story” with integration of necessary background.

The mechanism is highly novel in the bladder cancer domain. Pieces of the story are familiar beyond bladder cancer, but the identification of NETs in radioresistant cancer is novel. This also builds nicely on this group's prior work related to HMGB1 in radioresistance.

The main limitation of the manuscript is the lack of mechanistic details. It would be really intriguing to dissect out how NET formation is triggered (in addition to or through HMGB1) and especially how it imparts resistance to radiation.

In the clinical cohort, responders and non-responders appear quite balanced, but it would still be better to conduct a multivariable analysis to correct for potential confounders when looking at pre-treatment PMNs or NLR as prognostic markers.

The authors refer to synergy in Fig 3F – this needs to be tested formally.

Reviewer #2:

Remarks to the Author:

In this manuscript, Surasri et al. use in vivo and in vitro experiments to show that NETs promote resistance to radiation of bladder and lung cancer. Indeed, their data show that NETs are formed in the tumor microenvironment after radiation and that mice deficient in NETs or treated with NET-targeting strategies respond much better to radiotherapy. The authors propose that NETs can inhibit CD8 T cells (at least spatially) to counteract radiotherapy efficacy. Mechanistically, they identify HMGB1 as a source of NET-inducing factor in the tumor microenvironment.

Overall strengths: The manuscript is well written, the data are not over interpreted and do support their conclusions. To my knowledge, this is the first study showing that radiotherapy promotes NETs in the tumor microenvironment and that NETs can promote radiation resistance.

Overall weaknesses:

Key control experiments are lacking to conclude properly (as explained in the major and minor concerns below non-irradiated mice were not analyzed) and the methods used to detect NETs are not totally specific. The material and methods part is lacking important information. While the authors identify a negative correlation between NETs and intratumoral CD8 infiltration, they only propose possible mechanisms on how NETs could promote radio-resistance.

The work from Surasri et al. is of interest and represents another step toward the understanding of NETs in cancer resistance. However, the main issue I have for publication in nature communications is the lack of experimental controls and the lack of cellular and molecular mechanism, especially on how NETs can interfere with CD8 activity.

Major concerns:

1- The authors use CitH3 immunofluorescence to identify and quantify NETs in vivo. However, NETs are usually identified in vivo with the co-localization of CitH3 with one NET-associated proteases (such as Myeloperoxidase or Neutrophil Elastase). The quantification of NETs should take into account these parameters. Indeed, the method used by the authors do not distinguish between a NET or a cell positive for CitH3 (some Ly6G negative cells express CitH3 clearly in the representative pictures for example). In this context, it has recently been shown that cancer cells can secrete extracellular traps identified by CitH3 (Lai Shi et al, Mol Cancer Res, 2020). Therefore, there is a need to quantify NETs properly.

2- It is important to show the effect of radiation on non-tumor bearing tissues (normal bladder as a control for bladder cancer; normal lung as a control for lung cancer). This will help to understand if the recruitment of neutrophils and the formation of NET is dependent on the presence of the tumor.

3- In all the in vivo experiments, controls of mice without radiation are lacking.

4- In Fig. 2B, 2C, S1B, 3F and 3G it is important that the authors show the vehicle treatment for the administration of the NE inh. GLZ and DNase I. If these important controls were not used, it is important to repeat the experiment to reach any conclusion.

5- For all the in vivo experiments, the inhibitors (DNase, NE inh, GLZ) should be used without radiation to assess their role on tumor growth. Indeed, NETs can be detected in the tumor without radiation (Fig. 1D or data from other papers on NETs). Another important control is to assess the growth of tumor in WT and PAD4 KO mice without radiation. These important controls will help to understand whether the effect of NET inhibition is dependent on radiation.

6- The authors need to show and quantify the recruitment and the presence of NETs after irradiation of lung tumors as it has been done for the bladder cancer model.

7- The authors did not include a control in Fig. 3C (control medium). Moreover, the author would need to repeat the experiment using the MB-49 cell line which was used for the in vivo studies. In addition, other controls should be performed: effect of radiation of PMN alone and cancer cells alone (one can imagine that radiation-induced cell death is a source of extracellular DNA released in the medium, which might interfere with the analysis of the data. This is important as it has been shown that cancer cells can secrete extracellular traps; see Lai Shi et al, Mol Cancer Res, 2020)

8- The authors need to show and quantify the recruitment of neutrophils and inhibition of NET formation after GLZ treatment, NE inhibition and DNase I treatment in vivo. Indeed, it might help to explain whether the synergic effect of GLZ and DNase is due to incomplete inhibition of NETs by the single drugs. Another explanation is a role for intracellular HMGB1 (as shown by the authors previously), which needs to at least be discussed.

9- In Fig. 4A and 4B, the authors need to include representative images and quantification of the control, non-irradiated tumors. Moreover, it is surprising to see NETs at the tumor-stroma interface in Fig. 4A, while it did not appear to be the case in Fig 1. It is also important that the author explain how they were able to quantify CD8 T cells in or outside the tumor considering that the cancer cells are not identified/visualized in this experiment (through the expression of a fluorescent protein for example). If the tumor-stroma interface was visualized thanks to histological features, it needs to be explained properly.

10- It is unclear how NETs promote radio resistance. The data suggest that NETs counteract CD8 activity in the tumor microenvironment, but the cellular and molecular mechanisms regulating this process are missing. Multiple studies have shown that PMNs can inhibit CD8 T cells; therefore, it would be important to test whether the mechanisms attributed to CD8 inhibition by PMNs are NET-dependent.

Minor concerns:

1- Reference 13 indicates that neutrophils recruited in the TIME after radiation enhance the effect of radiotherapy. The authors should discuss the differences obtained between their study and ref 13.

2- The inability of PAD4 KO mice to form NETs after radiation need to be confirmed using immunofluorescence and the co-localization of Cit-H3 and myeloperoxidase or neutrophil elastase (as discussed in major concern 1)

3- The authors have previously identified intracellular functions of HMGB1 in radioresistance. From their current study, it is hard to analyze the effect of HMGB1 in the cancer cells versus the effect of extracellular HMGB1 on NET formation.

4- It is unclear why the authors switch from MB-49 cells used for their in vivo study to UM-UC3 cells for their in vitro study. Indeed, they showed previously that UM-UC3 express high levels of HMGB1, indicating that this cell line should induce NETs without the need of radiation. However, while UM-UC3 Conditioned media induce NETs (as imaged on figure 3D), it is not dependent on HMGB1 as GLZ has no effect.

5- The authors quantify NETs using % control in Fig. 1a and Arbitrary Unit in Fig. 1c. The author should use one or the other consistently.

6- The authors need to quantify the release of HMGB1 by non-irradiated and irradiated cancer cells.

7- In the legend of Fig. 3C, the authors indicate using MB-49 conditioned media, whereas the text and the figures indicate the use of UM-UC3 cells.

8- The authors state "co-culture of PMNs with irradiated UM-UC3 conditioned media ...". Co-culture is usually used for the culture between two different types of cells, not cells with conditioned media.

9- The authors need to cite Teijera et al (Immunity, 2020) when writing "... by forming a physical barrier between immune and tumor cells".

10- The authors need to show and quantify the recruitment of neutrophils and the presence of NETs in both the wild type versus the athymic mice model. Indeed, it is possible that the lack of CD4 and CD8 T cells influence the formation of NETs (even it doesn't seem to be the case, but controls are missing).

11- Considering that radiation should enhance neutrophils recruitment to the tumor, it would be interesting to analyze pre-RT versus post-RT PMN:CD8 ratio in responders versus non responders.

12- As the authors have access to the data, it is important to show the overall survival of patients with and without NETs post-RT.

13- It is unclear which type of patient is represented in Fig. 5G (non responders vs responders and pre-RT vs post-RT).

14- In the discussion, the authors state "we show that irradiation of urothelial cancer cells promotes NET formation in vitro, in vivo and in humans with MIBS". The authors have not shown that urothelial cancer cells promote NETs in human, but have shown a correlation, this needs to be corrected. I have the same concern when the authors state "this study is the first to report the NET inducing effect of radiation on a human cancer"; again, the authors are showing correlations.

15- The authors need to be more specific in the "in vivo tumor model" part of the material and methods of this manuscript. Indeed, it is important to indicate if the mice were anaesthetized during irradiation and whether control mice were also anaesthetized. Generally, the material and methods part is lacking specifics. For example, the catalog number is most of the time absent.

16- The authors need to discuss their results in the context of immunogenic radiotherapy, which have been shown to lead to an anti-tumor response (through the activation of cytotoxic T cells for example). Indeed, in their study radiotherapy leads to a protective effect of the cancer cells.

Reviewer #3:

Remarks to the Author:

De study by Shinde-Jadhav et al

TIME is part of the TME, several aspects of radio resistance affect both the immune status and radio resistance and must be seen as a whole. The paper is written from the perspective that all problems can be solved from the immune point of view thereby disregarding the more classical concepts of radio resistance that also find their roots in the tumor microenvironment and cellular characteristics (intrinsic resistance i.e. DNA dsb repair, accelerated proliferation, hypoxia responsible for HIF1a activation with consequential effects in the microenvironment). A main problem is the irradiation schedule used, why was it chosen, why fractionated and why not at least two dos levels. In the discussion the authors stress the need for a mechanistic answer for the observations, for this it is necessary to have the information mentioned above. Especially the fractionation makes it hard to find out the effects of irradiation since the second fraction seriously affects the T(I)ME.

Methods.

What was the reasoning behind the radiation schedule; the preferred schedule would be two doses, which allows to find dose effects, in a non-fractionated scheme, to prevent missing of kinetic effects and affecting/killing immune cells by the second fraction. These experiments are missing and need to be done.

TMA analysis: other markers relevant for radio resistance should be included.

Why did the investigators include a lung cancer cell line in their studies?

Results.

P7 the argument of UV in this perspective is unclear; the mode of action of UV is quite different from X-rays.

2x5Gy experiment fig 1: neutrophils already increase at 6h post irradiation at dose levels of 5-6Gy. In the design tumors are harvested at 72h after irradiation (second fraction?) the second fraction will have abolished effects from the first fraction. This makes interpretation hard. It seems from the images that there is a heterogeneous distribution of the green cells. Is this related to other microenvironmental factors such as hypoxia? One week after 2x5Gy necrosis could be a significant factor, was this assessed?

GD-experiments (fig 2) really should include two o levels, again fractionation complicate interpretation of the results. I Fig 2b 2c the controls (untreated) are missing, also in Fig 3. By leaving out a second dos level PLUS the controls it is difficult to interpret the effect of the interventions.

To assess if the observations are uniform or bladder cancer specific it is not enough to analyze one non-bladder tumor cell line

REVIEWER COMMENTS

Reviewer #1 (Remarks to the Author): with expertise in bladder cancer

The authors are investigating an important question – immune-related mechanisms of resistance to radiation therapy in bladder cancer. They have elegantly demonstrated that neutrophil extracellular traps contribute to resistance and suggest that they may accomplish this by building a barrier between infiltrating CD8+ T-cells and the tumor. They have used subcutaneous mouse models of bladder cancer and include some correlative human patient data to confirm the clinical relevance.

The paper is nicely written and is easy to read. The experiments are adequately described, and the results are clearly presented. The authors do a particularly good job of “telling the story” with integration of necessary background.

The mechanism is highly novel in the bladder cancer domain. Pieces of the story are familiar beyond bladder cancer, but the identification of NETs in radioresistant cancer is novel. This also builds nicely on this group’s prior work related to HMGB1 in radioresistance.

The main limitation of the manuscript is the lack of mechanistic details. It would be really intriguing to dissect out how NET formation is triggered (in addition to or through HMGB1) and especially how it imparts resistance to radiation.

- We thank the reviewer for this comment, and in our revised manuscript in terms of mechanism we have decided to explore how the radiated TIME triggers NET formation and promotes radioresistance. Our findings are in line with other studies suggesting that HMGB1 promotes NET formation through a TLR4 dependent mechanism. Indeed, we noted that PMNs from TLR4^{-/-} mice did not induce NETs when stimulated with recombinant HMGB1 or conditioned media from irradiated cancer cells. This is contrary to what we observed when human or murine PMNs were stimulated with rHMGB1 or irradiated conditioned media. We have now added these findings to Figure 3 E-G of our revised manuscript.

In the clinical cohort, responders and non-responders appear quite balanced, but it would still be better to conduct a multivariable analysis to correct for potential confounders when looking at pre-treatment PMNs or NLR as prognostic markers.

- We thank the reviewer for this suggestion. We performed a univariate and multivariable analysis on our cohort and have included these findings in Table 2 of our revised manuscript.
- In the univariate analysis, presence of NETs post-RT was associated with worse overall survival and this was also true when we adjusted for possible confounders in a multivariable analysis. In addition, a high PMN to CD8 ratio correlated with worse complete response and poor overall survival. In the multivariable analysis, it was still associated with complete response; however, this did not associate with overall survival. We believe this can be explained by a lack of statistical power and this does not

undermine our preclinical findings suggesting that pre-treatment PMNs can impact overall survival. Using a larger cohort of patients treated with RT from our center, we have observed that a high pre-treatment NLR is independently associated with worse overall survival and this is also the case when correcting for possible confounders (Supplemental Figure 6 – manuscript in preparation from this study).

The authors refer to synergy in Fig 3F – this needs to be tested formally.

- The reviewer has pointed out an important detail. Since we have not formally tested this in our study, we have reworded our findings in the revised manuscript.

Reviewer #2 (Remarks to the Author): with expertise in neutrophils/NETs

In this manuscript, Surashri et al. use in vivo and in vitro experiments to show that NETs promote resistance to radiation of bladder and lung cancer. Indeed, their data show that NETs are formed in the tumor microenvironment after radiation and that mice deficient in NETs or treated with NET-targeting strategies respond much better to radiotherapy. The authors propose that NETs can inhibit CD8 T cells (at least spatially) to counteract radiotherapy efficacy. Mechanistically, they identify HMGB1 as a source of NET-inducing factor in the tumor microenvironment.

Overall strengths: The manuscript is well written, the data are not over interpreted and do support their conclusions. To my knowledge, this is the first study showing that radiotherapy promote NETs in the tumor microenvironment and that NETs can promote radiation resistance.

Overall weaknesses:

Key control experiments are lacking to conclude properly (as explained in the major and minor concerns below non-irradiated mice were not analyzed) and the methods used to detect NETs are not totally specific. The material and methods part is lacking important information. While the authors identify a negative correlation between NETs and intratumoral CD8 infiltration, they only propose possible mechanisms on how NETs could promote radio-resistance.

Major concerns:

1- The authors use CitH3 immunofluorescence to identify and quantify NETs in vivo. However, NETs are usually identified in vivo with the co-localization of CitH3 with one NET-associated proteases (such as Myeloperoxidase or Neutrophil Elastase). The quantification of NETs should take into account these parameters. Indeed, the method used by the authors do not distinguish between a NET or a cell positive for CitH3 (some Ly6G negative cells express CitH3 clearly in the representative pictures for example). In this context, it has recently been shown that cancer cells can secrete extracellular traps identified by CitH3 (Lai Shi et al, Mol Cancer Res, 2020). Therefore, there is a need to quantify NETs properly.

- The reviewer has brought up an important point and in order to be also consistent with the literature, we have reformed and reanalyzed our immunofluorescence results by identifying NETs through positive staining for H3Cit and NE as suggested by the

reviewer. These results have been updated in the revised manuscript and figures: 1B, 1C, 4D, 5A, 5F, Figure 6, S1C, S2B.

2- It is important to show the effect of radiation on non-tumor bearing tissues (normal bladder as a control for bladder cancer; normal lung as a control for lung cancer). This will help to understand if the recruitment of neutrophils and the formation of NET is dependent on the presence of the tumor.

- The reviewer has brought up an important point and we have included these findings in Supplemental Figure 1C and have discussed our results in Figure 1 of the revised manuscript. In a normal murine bladder or irradiated bladder, no NE or H3Cit positive cells were observed. However, PMN infiltration was noted in tumor bearing bladders, and irradiated tumor-bearing bladders. NETs (identified through NE and H3Cit staining) were only noted in the irradiated tumor bearing bladders.

3- In all the *in vivo* experiments, controls of mice without radiation are lacking.

- Although we did not include these arms in our manuscript, all controls arms were performed for each experiment. These results have now been added to the figures of the revised manuscript (Figure 2B, Figure 4B)

4- In Fig. 2B, 2C, S1B, 3F and 3G it is important that the authors show the vehicle treatment for the administration of the NE inh. GLZ and DNase I. If these important controls were not used, it is important to repeat the experiment to reach any conclusion.

- All *in vivo* experiments were performed with the vehicle treatment for administration. We would like to thank the reviewer for pointing this detail and we have indicated this in our revised manuscript (S2C, S4)

5- For all the *in vivo* experiments, the inhibitors (DNase, NE inh, GLZ) should be used without radiation to assess their role on tumor growth. Indeed, NETs can be detected in the tumor without radiation (Fig. 1D or data from other papers on NETs). Another important control is to assess the growth of tumor in WT and PAD4 KO mice without radiation. These important controls will help to understand whether the effect of NET inhibition is dependent on radiation.

- These controls have now been added to our revised manuscript (Figure 2, Figure 4).

6- The authors need to show and quantify the recruitment and the presence of NETs after irradiation of lung tumors as it has been done for the bladder cancer model.

- We will be focusing our efforts by centering our manuscript on a bladder cancer model rather than introducing non-bladder cancer models as suggested by the reviewers and editor. As such, these results have been removed in our revised manuscript.

7- The authors did not include a control in Fig. 3C (control medium). Moreover, the author would need to repeat the experiment using the MB-49 cell line which was used for the *in vivo* studies. In addition, other controls should be performed: effect of radiation of PMN alone and cancer cells alone (one can imagine that radiation-induced cell death is a source of extracellular DNA released in the medium, which might interfere with the analysis of the data. This is

important as it has been shown that cancer cells can secrete extracellular traps; see Lai Shi et al, Mol Cancer Res, 2020)

- We performed these *in vitro* analyses using the human UM-UC3 cell line as we wanted to examine NET formation through stimulation of rHMGB1 or irradiated conditioned media in human PMNs. However, the reviewer brought up a valid point and we have repeated these findings using the murine MB49 cell line and bone-marrow derived PMNs from C57BL/6 mice (Figure 3). The reviewer has also brought up some important controls that were not included previously, so we have added these to our results as well. (Supplementary Figure 3).

8- The authors need to show and quantify the recruitment of neutrophils and inhibition of NET formation after GLZ treatment, NE inhibition and DNase I treatment *in vivo*. Indeed, it might help to explain whether the synergic effect of GLZ and DNase is due to incomplete inhibition of NETs by the single drugs. Another explanation is a role for intracellular HMGB1 (as shown by the authors previously), which needs to at least be discussed.

- We thank the reviewer for this suggestion, and we have performed immunofluorescence on these tissues and the results have been added to Figure 4 of our revised manuscript. We noted that while administering DNase I or GLZ alone decreases NET formation, this effect is pronounced when GLZ and DNase I are administered together.

9- In Fig. 4A and 4B, the authors need to include representative images and quantification of the control, non-irradiated tumors. Moreover, it is surprising to see NETs at the tumor-stroma interface in Fig. 4A, while it did not appear to be the case in Fig 1. It is also important that the author explain how they were able to quantify CD8 T cells in or outside the tumor considering that the cancer cells are not identified/visualized in this experiment (through the expression of a fluorescent protein for example). If the tumor-stroma interface was visualized thanks to histological features, it needs to be explained properly.

- We have now included the controls and quantified CD8 infiltration in these tissues in Figure 5. In regard to the tumor-stroma interface, these were visualized by histological features and this has now been explained in our methods section.

10- It is unclear how NETs promote radio resistance. The data suggest that NETs counteract CD8 activity in the tumor microenvironment, but the cellular and molecular mechanisms regulating this process are missing. Multiple studies have shown that PMNs can inhibit CD8 T cells; therefore, it would be important to test whether the mechanisms attributed to CD8 inhibition by PMNs are NET-dependent.

- We acknowledge that the mechanisms underlying CD8 inhibition by PMNs and NETs is certainly an interesting and valid question. In terms of mechanisms, there are two questions that arise 1) how does the radiated TIME trigger NET formation and promote radioresistance; 2) how NETs counteract CD8 T-cell activity. Since the novelty of our study is that radiation therapy induces NETs in the TIME, addressing the first question is highly relevant and we have now added to our results the role of TLR4 in our findings from Figure 3. Regarding the second question, understanding the downstream effects of NETs through its interactions with T-cells is a valid point; however, there is literature to support

that PMNs can suppress CD8 T-cells. Studies by Coffelt et al., 2015, Michaeli et al., 2017 and others have demonstrated that PMNs suppress T-cells in the context of cancer^{1, 2, 3, 4, 5}. Furthermore, this has also been attributed to NETs where recently the study by Tejeira et al., 2020 demonstrated that in the context of cancer, NETs can physically capture CD8 T-cells contributing to immunosuppression. This has also been noted in context of infection^{6, 7}. As such, although it is certainly an interesting point to explore, we believe it is beyond the scope of our current study and can be explored in subsequent studies.

Minor concerns:

1- Reference 13 indicates that neutrophils recruited in the TIME after radiation enhance the effect of radiotherapy. The authors should discuss the differences obtained between their study and ref 13.

- In this study, the authors have utilized a different cell line that has allowed them to monitor tumor growth post RT after depletion of PMNs for a longer window (~30days) than what we have observed in our study (~2weeks). In their study at 2 weeks, no difference is observed between PMN depletion in the RT arms vs RT+ aLy6G.

2- The inability of PAD4 KO mice to form NETs after radiation need to be confirmed using immunofluorescence and the co-localization of Cit-H3 and myeloperoxidase or neutrophil elastase (as discussed in major concern 1)

- We have now performed this analysis, and this has been added to Supplementary Figure 2B.

3- The authors have previously identified intracellular functions of HMGB1 in radioresistance. From their current study, it is hard to analyze the effect of HMGB1 in the cancer cells versus the effect of extracellular HMGB1 on NET formation.

- In our previous study (Ayoub et al., 2019) we have identified the role of extracellular HMGB1 in radioresistance and its effect on the immune landscape⁸. In our present study, we used GLZ as an inhibitor of HMGB1 to focus on the extracellular functions of HMGB1. We understand that this may not be clear in our text, so we have clarified this in our revised manuscript.

4- It is unclear why the authors switch from MB-49 cells used for their in vivo study to UM-UC3 cells for their in vitro study. Indeed, they showed previously that UM-UC3 express high levels of HMGB1, indicating that this cell line should induce NETs without the need of radiation. However, while UM-UC3 Conditioned media induce NETs (as imaged on figure 3D), it is not dependent on HMGB1 as GLZ has no effect.

- We used UM-UC3 as this is a human bladder cancer cell line and we wanted to examine NET induction in human PMNs. This will be clarified in the text in the revised manuscript. However, as the reviewer suggested in major concern 7, we have also repeated this experiment using the murine MB49 cell line and murine bone-marrow derived PMNs.

5- The authors quantify NETs using % control in Fig. 1a and Arbitrary Unit in Fig. 1c. The author should use one or the other consistently.

- We thank the reviewer for bringing up this detail and we have represented all these results (Figure 3) as fold control.

6- The authors need to quantify the release of HMGB1 by non-irradiated and irradiated cancer cells.

- We have performed this experiment in a previous publication (Ayoub et al., 2019) where we demonstrated that irradiation increased extracellular HMGB1 at a dose of 6Gy *in vitro*. We have now mentioned this in our text in the results section for Figure 4 of the revised manuscript.

7- In the legend of Fig. 3C, the authors indicate using MB-49 conditioned media, whereas the text and the figures indicates the use of UM-UC3 cells.

- We thank the reviewer for this point; it was a mistake from our side, and we have made this change in our revised manuscript

8- The authors state “co-culture of PMNs with irradiated UM-UC3 conditioned media ...”. Co-culture is usually used for the culture between two different types of cells, not cells with conditioned media.

- We thank the reviewer for noting this detail; we have rephrased this in our revised manuscript.

9- The authors need to cite Teijera et al (Immunity, 2020) when writing “... by forming a physical barrier between immune and tumor cells”.

- We have now made this change in our revised manuscript.

10- The authors need to show and quantify the recruitment of neutrophils and the presence of NETS in both the wild type versus the athymic mice model. Indeed, it is possible that the lack of CD4 and CD8 T cells influence the formation of NETs (even it doesn't seem to be the case, but controls are missing).

- We thank the reviewer for this suggestion, and we have now added this result to Figure 5. We did not observe any effect of the athymic mice model on PMN infiltration or NET formation.

11- Considering that radiation should enhance neutrophils recruitment to the tumor, it would be interesting to analyze pre-RT versus post-RT PMN:CD8 ratio in responders versus non responders.

- This was an interesting suggestion, and we performed the analysis to explore this. However, we did not observe any significant differences in the PMN:CD8 ratio between responders and non-responders in pre-RT compared to post-RT. We believe this may be due to the varying timepoints between biopsy obtained in post-RT patients or a lack of statistical power.

12- As the authors have access to the data, it is important to show the overall survival of patients with and without NETs post-RT.

- We thank the reviewer for this comment, and we have included this in our revised manuscript in Supplemental Figure 5.

13- It is unclear which type of patient is represented in Fig. 5G (non responders vs responders and pre-RT vs post-RT).

- This was a representative picture from a non-responder post-RT, and we have now clarified this in our revised manuscript in the figure legend.

14- In the discussion, the authors state “we show that irradiation of urothelial cancer cells promotes NET formation in vitro, in vivo and in humans with MIBS”. The authors have not shown that urothelial cancer cells promote NETs in human, but have shown a correlation, this needs to be corrected. I have the same concern when the authors state “this study is the first to report the NET inducing effect of radiation on a human cancer”; again, the authors are showing correlations.

- We have now made these changes in the revised manuscript discussion.

15- The authors need to be more specific in the “in vivo tumor model” part of the material and methods of this manuscript. Indeed, it is important to indicate if the mice were anaesthetized during irradiation and whether control mice were also anaesthetized. Generally, the material and methods part is lacking specifics. For example, the catalog number is most of the time absent.

- As suggested, we have added more details regarding the *in vivo* tumor model in the material and methods section in our revised manuscript.

16- The authors need to discuss their results in the context of immunogenic radiotherapy, which have been shown to lead to an anti-tumor response (through the activation of cytotoxic T cells for example). Indeed, in their study radiotherapy leads to a protective effect of the cancer cells.

- We thank the reviewer for bringing up this point and we have added this to our discussion in the revised manuscript.

Reviewer #3 (Remarks to the Author): with expertise in radio-resistance

De study by Shinde-Jadhav et al

TIME is part of the TME, several aspects of radio resistance affect both the immune status and radio resistance and must be seen as a whole. The paper is written from the perspective that all problems can be solved from the immune point of view thereby disregarding the more classical concepts of radio resistance that also find their roots in the tumor microenvironment and cellular characteristics (intrinsic resistance i.e. DNA dsb repair, accelerated proliferation, hypoxia responsible for HIF1a activation with consequential effects in the microenvironment). A

main problem is the irradiation schedule used, why was it chosen, why fractionated and why not at least two dose levels. In the discussion the authors stress the need for a mechanistic answer for the observations, for this it is necessary to have the information mentioned above. Especially the fractionation makes it hard to find out the effects of irradiation since the second fraction seriously affects the T(I)ME.

Methods.

What was the reasoning behind the radiation schedule; the preferred schedule would be two doses, which allows to find dose effects, in a non-fractionated scheme, to prevent missing of kinetic effects and affecting/killing immune cells by the second fraction. These experiments are missing and need to be done.

- We would like to thank the reviewer for their valuable comments about the radiation schedule used in our manuscript. In our study, we chose a fractionated radiation schedule to mimic what is performed in the clinical setting. In our previous study, (Ayoub et al.,2019) we utilized the same fractionated radiation schedule with the cell line MB49 and observed no significant differences in the TIME of control or irradiated mice through flow cytometry.
- The reviewer has brought up an important point regarding the dose-effects, and we addressed this in Figure 1. We explored the dose-effect of irradiation on NET formation through the following experiment: C57BL/6 mice were injected with MB49 cells, when tumors were palpable, they were irradiated with a single low dose (2Gy) or single high dose (10Gy) of radiation. Tumors were collected for immunofluorescence analysis to evaluate PMN infiltration and presence of NETs in the TIME when tumors were irradiated in a non-fractionated scheme. We observed a dose response in PMN infiltration where radiation increase PMN infiltration in tumors radiated with 2x5Gy and 10Gy at 72hrs post-RT and 1-week post-RT. In addition, a dose response increase in NETs was observed. These results have been discussed in Figure 1.

TMA analysis: other markers relevant for radio resistance should be included.

- While this is an interesting point, we believe that it is beyond the scope of our study. Our aim with the TMA analysis was to complement our preclinical findings; however performing this analysis can be interesting for a subsequent study.

Why did the investigators include a lung cancer cell line in their studies?

- In the revised manuscript, we will be focusing our findings in the bladder tumor model only as suggested by the editor.

Results.

P7 the argument of UV in this perspective is unclear; the mode of action of UV is quite different from X-rays.

2x5Gy experiment fig 1: neutrophils already increase at 6h post irradiation at dose levels of 5-6Gy. In the design tumors are harvested at 72h after irradiation (second fraction?) the second fraction will have abolished effects from the first fraction. This makes interpretation hard. It

seems from the images that there is a heterogeneous distribution of the green cells. Is this related to other microenvironmental factors such as hypoxia? One week after 2x5Gy necrosis could be a significant factor, was this assessed?

GD-experiments (fig 2) really should include two o levels, again fractionation complicate interpretation of the results. I Fig 2b 2c the controls (untreated) are missing, also in Fig 3. By leaving out a second dose level PLUS the controls it is difficult to interpret the effect of the interventions.

- We thank the reviewer for this point; we performed H&E on the tumors and included these images in Supplementary Figure 1B. When taking the confocal images, tumor regions were identified through H&E and confirmed the DAPI staining which has been added to our methods section. MB49 tumors irradiated at 10Gy showed increased necrosis 1-week post-RT than those that were treated with 2x5Gy or 2Gy. Since necrosis and ulceration in our tumor model would bias our results and findings, we chose to perform our in vivo studies with a dose of 2x5Gy, in addition to the reasons mentioned above.

To assess if the observations are uniform or bladder cancer specific it is not enough to analyze one non-bladder tumor cell line

- We will be focusing our efforts by centering our manuscript on a bladder cancer model rather than introducing non-bladder cancer models as suggested by the editor.

References:

1. Coffelt SB, *et al.* IL-17-producing gammadelta T cells and neutrophils conspire to promote breast cancer metastasis. *Nature* **522**, 345-348 (2015).
2. Kalyan S, Kabelitz D. When neutrophils meet T cells: beginnings of a tumultuous relationship with underappreciated potential. *Eur J Immunol* **44**, 627-633 (2014).
3. Lelifeld PH, Koenderman L, Pillay J. How Neutrophils Shape Adaptive Immune Responses. *Front Immunol* **6**, 471 (2015).
4. Michaeli J, *et al.* Tumor-associated neutrophils induce apoptosis of non-activated CD8 T-cells in a TNFalpha and NO-dependent mechanism, promoting a tumor-supportive environment. *Oncoimmunology* **6**, e1356965 (2017).
5. Singel KL, *et al.* Mature neutrophils suppress T cell immunity in ovarian cancer microenvironment. *JCI Insight* **4**, (2019).
6. Teijeira A, *et al.* CXCR1 and CXCR2 Chemokine Receptor Agonists Produced by Tumors Induce Neutrophil Extracellular Traps that Interfere with Immune Cytotoxicity. *Immunity* **52**, 856-871 e858 (2020).
7. Sivanandham R, *et al.* Neutrophil extracellular trap production contributes to pathogenesis in SIV-infected nonhuman primates. *J Clin Invest* **128**, 5178-5183 (2018).
8. Ayoub M, *et al.* The immune mediated role of extracellular HMGB1 in a heterotopic model of bladder cancer radioresistance. *Sci Rep* **9**, 6348 (2019).

Reviewers' Comments:

Reviewer #1:

Remarks to the Author:

The authors have addressed reviewer comments carefully and satisfactorily.

Reviewer #2:

Remarks to the Author:

I would like to thank the authors for their work and for answering all my concerns. This manuscript of high quality show how RT-induced NET formation promote radioresistance and this is an important point for the cancer biology field and neutrophil biology field.

Reviewer #3:

Remarks to the Author:

Substantial part of my comments were addressed

REVIEWERS' COMMENTS

Reviewer #1 (Remarks to the Author):

The authors have addressed reviewer comments carefully and satisfactorily.

Reviewer #2 (Remarks to the Author):

I would like to thank the authors for their work and for answering all my concerns. This manuscript of high quality show how RT-induced NET formation promote radioresistance and this is an important point for the cancer biology field and neutrophil biology field.

Reviewer #3 (Remarks to the Author):

Substantial part of my comments were addressed

We would like to thank the reviewers for their comments and suggestions.